# Delving into Muon and Beyond: Deep Analysis and Extensions

Xianbiao Qi [* 1]  Marco Chen [* 2]  Jiaquan Ye [1]  Yelin He [1]  Rong Xiao [† 1]

## Abstract

The Muon optimizer has recently attracted considerable attention for its strong empirical performance and use of orthogonalized updates on matrix-shaped parameters, yet its underlying mechanisms and relationship to adaptive optimizers such as Adam remain insufficiently understood. In this work, we aim to address these questions through a unified spectral perspective. Specifically, we view Muon as the $p = 0$ endpoint of a family of spectral transformations of the form $U\Sigma^p V^\top$, and consider additional variants with $p = \frac{1}{2}$, $p = \frac{1}{4}$, and $p = 1$. These transformations are applied to both first-moment updates, as in momentum SGD, and to root-mean-square (RMS) normalized gradient updates as in Adam. To enable efficient computation, we develop a coupled Newton iteration that avoids explicit singular value decomposition. Across controlled experiments, we find that RMS-normalized updates yield more stable optimization than first-moment updates. Moreover, while spectral compression provides strong stabilization benefits under first-moment updates, the Muon update ($p = 0$) does not consistently outperform Adam. These results suggest that Muon is best understood as an effective form of spectral normalization, but not a universally superior optimization method. Our code is available at https://github.com/Ocram7/BeyondMuon.

## 1. Introduction

Optimization methods (Robbins & Monro, 1951; Nesterov, 1983; Duchi et al., 2011; Hinton et al., 2012; Kingma & Ba, 2014; Loshchilov & Hutter, 2019; Gupta et al., 2018; Jordan et al., 2024) are a cornerstone of modern deep learning.

*Equal contribution [1]Intellifusion Inc. [2]Tsinghua University, marco.ty.chen@gmail.com. Correspondence to: Rong Xiao <rongxiao@gmail.com>.

*Proceedings of the 43$^{rd}$ International Conference on Machine Learning*, Seoul, South Korea. PMLR 306, 2026. Copyright 2026 by the author(s).

Among them, Adam (Kingma & Ba, 2014) and its variants (Loshchilov & Hutter, 2019; Shazeer & Stern, 2018) are widely adopted due to their strong empirical stability and robustness across architectures and scales. By combining momentum with variance-based normalization, Adam incorporates second-moment information while remaining computationally efficient, and has consequently become the default optimizer in large-scale neural network training (Touvron et al., 2023a;b; Dubey et al., 2024; Guo et al., 2025; Yang et al., 2025).

Recently, Muon (Jordan et al., 2024) has attracted growing attention as an alternative optimization strategy, particularly in large language models (Team et al., 2025). Unlike most conventional optimizers that operate element-wise, Muon applies matrix-level transformations to gradients, enforcing orthogonality through a spectral operation. Empirical evidence suggests that Muon can improve convergence behavior, leading to growing interest in the community. Despite growing adoption, the effectiveness and role of Muon remain insufficiently understood. Existing work has either focused on empirical performance gains (Jordan et al., 2024; Team et al., 2025) or on mathematical analysis (Newhouse, 2025; Bernstein, 2025a; Buchanan, 2025), but these studies are often accompanied by additional techniques such as QK-Norm or QK-Clip (Team et al., 2025). Consequently, it remains unclear to what extent the reported improvements can be attributed to Muon itself, how Muon relates to established adaptive optimizers such as Adam, and whether Muon can be systematically improved. These ambiguities motivate a careful and controlled re-examination of Muon.

In this work, our goal is to study the effectiveness of the Muon optimizer through a unified spectral framework and controlled empirical evaluation. We view Muon as the $p = 0$ endpoint of a family of spectral transformations of the form $U\Sigma^p V^\top$, and consider fractional variants with $p = \frac{1}{2}$ and $p = \frac{1}{4}$, as well as $p = 1$, which recovers a standard gradient update. We evaluate these transformations under both first-moment updates, as in momentum SGD, and root-mean-square-normalized updates, as in Adam. To make fractional spectral updates practical, we introduce a coupled Newton-Schulz iteration method that computes $U\Sigma^{\frac{1}{2}}V^\top$ and $U\Sigma^{\frac{1}{4}}V^\top$ using only matrix multiplications, avoiding explicit SVD.

To ensure fair and interpretable comparisons, we design our experiments to be highly controlled. We explicitly decouple matrix and vector learning rates for all methods, and avoid weight decay as well as auxiliary stabilizing techniques like QK-Norm and QK-Clip in the main experiments. Later, we ablate weight decay and QK-Norm to examine whether the trends from our primary experiments persist under more practical settings.

Under these settings, our experiments yield three key findings: (i) Muon ($p = 0$) is crucial for stabilizing first-moment updates and substantially improves robustness over mSGD; (ii) when applied to RMS-normalized updates, Muon-like orthogonalization does not consistently outperform Adam and is matched or exceeded by partial spectral compression; and (iii) across spectral variants, RMS-normalized variants are generally stronger than their first-moment counterparts.

Our main contributions are summarized as follows:

- We introduce a unified spectral framework $\boldsymbol{U}\boldsymbol{\Sigma}^p\boldsymbol{V}^\top$ that places Muon as the $p = 0$ endpoint and propose intermediate variants with $p = \frac{1}{2}$ and $p = \frac{1}{4}$.

- We develop a coupled Newton–Schulz iteration method that enables efficient computation of the fractional spectral updates $p = \frac{1}{2}$ and $p = \frac{1}{4}$ without explicit SVD, and analyze its approximation quality and runtime overhead.

- We provide a controlled empirical comparison between Muon, Adam, and their spectral variants across first-moment and RMS-normalized updates, isolating spectral effects from other confounding techniques.

**Conflict of Interest Disclosure.** The authors declare no financial interests beyond the disclosed affiliations.

## 2. Related Work

**Optimizers before Transformer.** Before the Transformer (Vaswani et al., 2017), CNNs (LeCun et al., 2002; He et al., 2016) and shallow recurrent models such as LSTMs (Hochreiter & Schmidhuber, 1997) dominated neural network design. These architectures were typically of moderate depth with limited cumulative nonlinearity, resulting in relatively small variation in layer-wise Lipschitz constants (Qi et al., 2023a;b). Consequently, training with a global learning rate was generally effective. Under this regime, classical stochastic optimization methods (Bottou et al., 2018) were sufficient. Stochastic Gradient Descent (SGD) (Robbins & Monro, 1951) and its variants (Nesterov, 1983; Johnson & Zhang, 2013), often combined with simple learning rate schedules, were the standard choices.

Although adaptive methods such as Adagrad (Duchi et al., 2011), RMSProp (Hinton et al., 2012), and Adam (Kingma & Ba, 2014) had been proposed, they were not widely adopted at the time. This can be attributed to two main factors. First, weight decay was primarily viewed as a regularization technique, and its interaction with adaptive optimizers was poorly understood; in particular, coupling weight decay with gradients rather than weights significantly degraded the performance of Adam-type methods (Loshchilov & Hutter, 2019). Second, in CNN-based models, inter-layer differences in Lipschitz constants were insufficient to necessitate per-parameter adaptivity, and momentum SGD already achieved strong empirical performance. As a result, Adam did not consistently outperform mSGD in this setting.

**Optimizers after Transformer.** The introduction of Transformers (Vaswani et al., 2017) fundamentally altered the optimization landscape of deep learning. Compared to earlier architectures, Transformer-based models are substantially deeper and exhibit stronger nonlinearities (Shazeer, 2020), arising from attention mechanisms (Vaswani et al., 2017), residual connections (He et al., 2016), and normalization layers (Ba et al., 2016). Consequently, different layers and parameter groups often exhibit markedly different curvature and gradient statistics, often leading to training instability and suboptimal convergence.

AdamW (Loshchilov & Hutter, 2019) resolved the improper coupling between weight decay and adaptive learning rates in Adam, leading to significantly improved performance on Transformer models. As a result, AdamW consistently outperforms momentum SGD in large-scale Transformer training and has become a standard optimizer choice. Subsequent work has proposed a range of alternative optimizers, including Adafactor (Shazeer & Stern, 2018), SignSGD (Bernstein et al., 2018), LAMB (You et al., 2019), Adan (Xie et al., 2024), Lion (Chen et al., 2023), Sophia (Liu et al., 2024), Mars (Yuan et al., 2025), and related variants (Zhang et al., 2025; Liang et al., 2025). These methods primarily target improved stability, efficiency, or convergence in large-scale, highly nonlinear training regimes.

**Matrix-based Optimizers.** Matrix-based optimizers exploit the structured geometry of parameters by operating directly on matrix-valued variables, rather than treating parameters as independent scalars. Early work in this direction focused on making second-order optimization tractable via structured approximations. K-FAC (Martens & Grosse, 2015) introduced a Kronecker-factored approximation of the Fisher information matrix, enabling efficient layer-wise preconditioning, while Shampoo (Gupta et al., 2018) extended this idea by applying Kronecker factorizations along multiple matrix dimensions, leading to improved stability. These methods share the goal of incorporating curvature information through structured matrix approximations.

More recently, Muon (Jordan et al., 2024) has emerged as a matrix-based optimizer that departs from explicit cur-

vature modeling. Muon directly updates matrix-valued parameters via orthogonalization and spectral transformations computed using Newton–Schulz iterations (Schulz, 1933; Higham, 2008), relying solely on matrix multiplications. Since its introduction, Muon has inspired a growing body of follow-up work (Team et al., 2025; Si et al., 2025; Frans et al., 2025; Su, 2025; Lau et al., 2025), including analyses that clarify its numerical linear algebra foundations (Bernstein, 2025a; Newhouse, 2025) and geometric interpretations that view it as implicit manifold-aware optimization (Bernstein, 2025b; Buchanan, 2025).

**Remarks.** Unlike several previous works on optimizer benchmarking (Wen et al., 2025; Frans et al., 2025), our study is characterized by the following features: (1) we introduce a unified spectral perspective $U\Sigma^p V^\top$ with $p \in [0, 1]$, and instantiate it with four concrete choices, $p \in \{1, \frac{1}{2}, \frac{1}{4}, 0\}$. For $U\Sigma^{\frac{1}{2}}V^\top$ and $U\Sigma^{\frac{1}{4}}V^\top$, we develop a coupled Newton–Schulz algorithm to solve them. (2) we experimentally show that matrix-based optimizers with second-moment information are stronger than their pure first-moment counterparts. (3) we design our experiments to be highly controlled. We explicitly decouple matrix and vector learning rates for all methods, disable weight decay, and avoid auxiliary techniques. This allows us to isolate the intrinsic capabilities of each optimizer, enabling a fair and transparent comparison.

## 3. Delving into Muon and Beyond

### 3.1. Baselines: Adam and Muon

As in Muon, our study focuses exclusively on *matrix-shaped parameters* $W \in \mathbb{R}^{m \times n}$; all vector parameters are optimized using standard Adam. In this subsection, we briefly review Adam and Muon and fix notation.

**Adam.** Adam (Kingma & Ba, 2014) is a popular stochastic optimizer that combines momentum with adaptive, second-moment-based normalization. Given the gradient matrix $G_t = \nabla_{W_t}\ell(W_t)$ at iteration $t$, Adam maintains exponentially decaying estimates of the first and second moments:

$$M_t = \beta_1 M_{t-1} + (1 - \beta_1)G_t, \quad (1)$$

$$V_t = \beta_2 V_{t-1} + (1 - \beta_2)G_t^2, \quad (2)$$

and updates parameters via

$$W_{t+1} = W_t - \eta_t M_t \oslash \sqrt{V_t}. \quad (3)$$

Here, $\beta_1$ and $\beta_2$ are the first- and second-moment decay rates, $\eta_t$ is the learning rate, and $\oslash$ denotes elementwise division. From a preconditioning perspective, Adam applies a diagonal matrix $V_t^{-1/2}$ to the momentum $M_t$, normalizing each coordinate by an estimate of its second-moment

and yielding bounded, scale-adaptive updates. For brevity, throughout the rest of this work we refer to $M_t \oslash \sqrt{V_t}$ as the *RMS-normalized* update, since $V_t$ tracks an exponential moving average of squared gradients.

**Muon.** Muon (Jordan et al., 2024) replaces Adam's elementwise normalization with a *spectral normalization* of the update matrix. Given the momentum matrix $M_t$ with singular value decomposition $M_t = U\Sigma V^\top$, Muon defines the update direction as the polar factor

$$\mathcal{P}(M_t) := UV^\top,$$

which discards the singular values and retains only the left and right singular vectors. Computing the polar factor is typically referred to as matrix orthogonalization.

From an optimization perspective, orthogonalization enforces *unit magnitude along every singular direction*. In contrast to Adam, which normalizes updates on an elementwise basis via a diagonal preconditioner, Muon equalizes the strength of updates across all directions of the gradient matrix. As a result, Muon removes anisotropic scaling in the spectrum of $M_t$, yielding directionally balanced but magnitude-agnostic updates.

**Newton–Schulz approximation.** Computing the polar factor $\mathcal{P}(M_t) = UV^\top = M_t(M_t^\top M_t)^{-1/2}$ via an explicit SVD is prohibitively expensive for large models. However, we can approximate the inverse square root using *Newton–Schulz iteration*, which relies only on matrix multiplications. Let $A = \alpha M_t^\top M_t$, where the scaling constant $\alpha$ is chosen such that $\|I - A\|_2 \leq 1$. The iteration

$$Z_{k+1} = \tfrac{1}{2} Z_k(3I - AZ_k^2), \quad Z_0 = I, \quad (4)$$

converges quadratically to $\frac{1}{\sqrt{\alpha}}(M_t^\top M_t)^{-1/2}$. After $K$ iterations, the polar factor is approximated as

$$\mathcal{P}(M_t) \approx \sqrt{\alpha}M_t Z_K. \quad (5)$$

In practice, Muon (Jordan et al., 2024) employs Algorithm 2 in Appendix B, which is a slightly more direct and aggressive version of the Newton-Schulz iteration shown in Equation 4. The official implementation directly computes the polar factor with a procedure that converges for initializations where the singular values are adequately normalized.

### 3.2. Momentum vs. RMS-Normalized Updates

**Where to apply spectral transforms.** Muon (Jordan et al., 2024) applies its spectral operation to the *first-moment* momentum $M_t$. However, in large-scale Transformer training, Adam-style optimizers that incorporate second-moment statistics generally outperform momentum SGD-based optimizers that only leverage first-moment information. This

observation motivates us to examine whether spectral transformations yield additional benefits when applied to rms-normalized updates. Concretely, we study the two inputs

$$\boldsymbol{O}_t^{\mathrm{mom}} := \boldsymbol{M}_t \quad \text{and} \quad \boldsymbol{O}_t^{\mathrm{rms}} := \boldsymbol{M}_t \oslash \sqrt{\boldsymbol{V}_t},$$

which correspond to the first-moment update shown in Equation 1 and the rms-normalized update from Equation 3.

**How $\boldsymbol{O}_t^{\mathrm{mom}}$ and $\boldsymbol{O}_t^{\mathrm{rms}}$ differ.** The key distinction between these two regimes lies in how update magnitudes are controlled. In first-order momentum methods, the update $\boldsymbol{M}_t$ aggregates gradients over time without explicit normalization by their scale. As a result, the norm of the update can grow with accumulated gradient magnitude, especially under high variance or anisotropic curvature. Consequently, the update scale is not bounded by the optimizer.

In contrast, RMS-normalized updates explicitly rescale the momentum using second-moment statistics. The update $\boldsymbol{M}_t \oslash \sqrt{\boldsymbol{V}_t}$ normalizes each coordinate by an estimate of its raw second moment, yielding an update whose magnitude is provably bounded. For Adam-style methods, this bound, $\frac{1-\beta_1}{\sqrt{1-\beta_2}}$ (Qi et al., 2023b), depends only on the decay rates and is independent of the raw gradient magnitude. This intrinsic normalization leads to improved numerical stability and more predictable optimization dynamics.

### 3.3. A Spectral Family of Transformations

We introduce a family of spectral gradient transformations that enables a unified analysis of Muon-inspired spectral methods. Given the SVD of $\boldsymbol{O}_t$ as $\boldsymbol{O}_t = \boldsymbol{U}\boldsymbol{\Sigma}\boldsymbol{V}^\top$, we define the general spectral transformation

$$\Psi_p(\boldsymbol{O}_t) := \boldsymbol{U}\boldsymbol{\Sigma}^p \boldsymbol{V}^\top, \qquad p \in [0, 1]. \qquad (6)$$

The exponent $p$ directly controls how the singular spectrum is rescaled. Writing the SVD in its rank-one form, we obtain

$$\Psi_p(\boldsymbol{O}_t) = \sum_{i=1}^d \sigma_i^p \, \boldsymbol{u}_i \boldsymbol{v}_i^\top, \qquad (7)$$

which makes it explicit that decreasing $p$ progressively compresses the singular values $\{\sigma_i\}$ while preserving the singular directions $\{\boldsymbol{u}_i, \boldsymbol{v}_i\}$.

This formulation subsumes several existing methods and enables systematic comparisons across spectral behaviors. In particular, $p = 1$ leaves the input unchanged, recovering standard methods like mSGD and Adam, while $p = 0$ maps all nonzero singular values to 1, yielding an orthogonalized update direction. When the input is the first-moment momentum $\boldsymbol{O}_t = \boldsymbol{O}_t^{\mathrm{mom}}$, $p = 0$ corresponds to Muon.

### 3.4. Instantiating the Spectral Family

**From a spectral operator to concrete optimizers.** We instantiate our spectral update family by applying the map

$\Psi_p(\cdot)$ from Equation 6 to one of two matrix-valued inputs: the first-moment momentum $\boldsymbol{O}_t^{\mathrm{mom}}$, or the RMS-normalized update $\boldsymbol{O}_t^{\mathrm{rms}}$. We consider four spectral exponents

$$p \in \left\{1, \tfrac{1}{2}, \tfrac{1}{4}, 0\right\},$$

where $p = 1$ leaves the input unchanged and $p = 0$ collapses the spectrum to the polar factor. The intermediate powers $p = \frac{1}{2}$ and $p = \frac{1}{4}$ provide milder alternatives to Muon's fully flattened spectrum and can be computed efficiently via coupled Newton–Schulz iteration, as described in Subsection 3.5. This yields the controlled interpolation

$$\boldsymbol{U}\boldsymbol{\Sigma}^1\boldsymbol{V}^\top \;\to\; \boldsymbol{U}\boldsymbol{\Sigma}^{1/2}\boldsymbol{V}^\top \;\to\; \boldsymbol{U}\boldsymbol{\Sigma}^{1/4}\boldsymbol{V}^\top \;\to\; \boldsymbol{U}\boldsymbol{\Sigma}^0\boldsymbol{V}^\top.$$

Combining two inputs with four exponents produces the eight optimizer instances evaluated in this work.

**Naming convention.** Each instance is identified by its *input family* and *spectral exponent*. We write

$$\textsc{BaseX} \;:\; \Delta\boldsymbol{W}_t \propto \Psi_p(\boldsymbol{O}_t),$$

where $\textsc{Base} \in \{\textbf{mSGD}, \textbf{Adam}\}$ selects $\boldsymbol{O}_t^{\mathrm{mom}}$ or $\boldsymbol{O}_t^{\mathrm{rms}}$, and the suffix $\text{X} \in \{\varnothing, \textbf{S}, \textbf{Q}, \textbf{Z}\}$ encodes the exponent

$\varnothing \leftrightarrow p = 1$ (identity), $\qquad \textbf{S} \leftrightarrow p = \frac{1}{2}$ (square-root),
$\textbf{Q} \leftrightarrow p = \frac{1}{4}$ (quarter-power), $\quad \textbf{Z} \leftrightarrow p = 0$ (zero-power).

**Momentum-input family ($\boldsymbol{O}_t^{\mathrm{mom}} = \boldsymbol{M}_t$).** Under the momentum input, the four instances are

$$\textbf{mSGD}, \; \textbf{mSGDS}, \; \textbf{mSGDQ}, \; \textbf{mSGDZ},$$

corresponding to $p \in \{1, \frac{1}{2}, \frac{1}{4}, 0\}$ respectively. The endpoint **mSGDZ** is equivalent to **Muon** since $\Psi_0(\boldsymbol{M}_t) = \mathcal{P}(\boldsymbol{M}_t) = \boldsymbol{U}\boldsymbol{V}^\top$.

**RMS-normalized-input family ($\boldsymbol{O}_t^{\mathrm{rms}} = \boldsymbol{M}_t \oslash \sqrt{\boldsymbol{V}_t}$).** Under the RMS-normalized input, the four instances are

$$\textbf{Adam}, \; \textbf{AdamS}, \; \textbf{AdamQ}, \; \textbf{AdamZ},$$

again corresponding to $p \in \{1, \frac{1}{2}, \frac{1}{4}, 0\}$ respectively.

**Spectral anisotropy control.** The four spectral variants in each input family share the same singular vectors $(\boldsymbol{U}, \boldsymbol{V})$ and differ only in how the singular values are rescaled. Thus, the effect of the spectral transformation can be understood as simply reshaping the anisotropy of the singular spectrum. To quantify anisotropy, we use the spectral condition number over the nonzero singular spectrum. Given the largest and smallest nonzero singular values of $\boldsymbol{O}_t$, denoted by $\sigma_{\max}$ and $\sigma_{\min}$, respectively, we define the condition number as

$$\kappa(\boldsymbol{O}_t) = \frac{\sigma_{\max}}{\sigma_{\min}}.$$

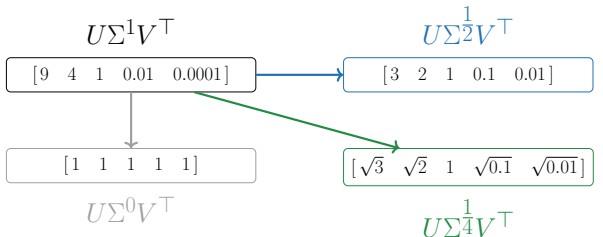

*Figure 1.* Effect of spectral exponent $p$ on the singular spectrum (illustrative numbers). Decreasing $p$ compresses the spectrum: large singular values are damped relative to small ones, and $p = 0$ maps all nonzero singular values to 1.

Intuitively, $\kappa(\boldsymbol{O}_t)$ measures the spread of the singular values. If the input is highly ill-conditioned and a small number of dominant singular directions disproportionately influence the update, $\kappa(\boldsymbol{O}_t)$ is large.

If the input has singular values $\{\sigma_i\}_{i=1}^d$, then $\Psi_p$ produces singular values $\{\sigma_i^p\}_{i=1}^d$, so the condition number becomes

$$\kappa_p \;=\; \frac{\sigma_{\max}^p}{\sigma_{\min}^p} \;=\; \kappa(\boldsymbol{O}_t)^p.$$

Therefore, decreasing $p$ reduces spectral anisotropy:

$$\kappa_1 \geq \kappa_{1/2} \geq \kappa_{1/4} \geq \kappa_0 = 1,$$

Indeed, for the toy spectrum in Figure 1, $\kappa_1 = 90000$, $\kappa_{1/2} = 300$, $\kappa_{1/4} = \sqrt{300}$, and $\kappa_0 = 1$. This viewpoint clarifies how the exponent $p$ continuously interpolates between the original spectrum ($p = 1$) and complete spectral flattening ($p = 0$).

**The effect of spectral compression.** As shown in Figure 1, decreasing $p$ progressively compresses the *entire* singular-value spectrum. Hence, directions associated with large singular values are attenuated, while those associated with small singular values (less than one) are amplified.

### 3.5. Efficient Computation

**Computing $\Psi_{1/2}(\boldsymbol{O}_t)$ and $\Psi_{1/4}(\boldsymbol{O}_t)$.** A practical challenge in spectral methods is computational efficiency, as direct SVD computation is infeasible for large models. Fortunately, we can rewrite $\Psi_{1/2}(\boldsymbol{O}_t)$ in a form that avoids explicit SVD. Let $\boldsymbol{O}_t = \boldsymbol{U}\boldsymbol{\Sigma}\boldsymbol{V}^\top \in \mathbb{R}^{m \times n}$. Then,

$$\Psi_{1/2}(\boldsymbol{O}_t) = \boldsymbol{U}\boldsymbol{\Sigma}^{1/2}\boldsymbol{V}^\top \;=\; \boldsymbol{O}_t\,(\boldsymbol{O}_t^\top\boldsymbol{O}_t)^{-1/4}.$$

Thus, it suffices to compute an inverse fourth root of the symmetric matrix $\boldsymbol{X} := \boldsymbol{O}_t^\top\boldsymbol{O}_t \in \mathbb{R}^{n \times n}$. We can compute $\boldsymbol{X}^{\frac{1}{2}}$ and $\boldsymbol{X}^{\frac{-1}{2}}$ efficiently using the coupled Newton-Schulz algorithm discussed below, and obtain $\boldsymbol{X}^{-1/4}$ by applying the same procedure to $\boldsymbol{X}^{1/2}$ (i.e., computing $(\boldsymbol{X}^{1/2})^{-1/2} = \boldsymbol{X}^{-1/4}$). Consequently, $\Psi_{1/2}(\boldsymbol{O})$ can be implemented using only matrix multiplications as $\Psi_{1/2}(\boldsymbol{O}) \;=\; \boldsymbol{O}\,\boldsymbol{X}^{-1/4}$.

---

**Algorithm 1** Coupled Newton-Schulz for $\boldsymbol{X}^{\frac{1}{2}}$ and $\boldsymbol{X}^{\frac{-1}{2}}$

**Require:** $\boldsymbol{X}$, number of iterations $K$
**Ensure:** $\boldsymbol{X}^{\frac{1}{2}}$, $\boldsymbol{X}^{-\frac{1}{2}}$
1: $\boldsymbol{Y}_0 \leftarrow \boldsymbol{X}, \quad \boldsymbol{Z}_0 \leftarrow \boldsymbol{I}$
2: $\alpha \leftarrow \|\boldsymbol{X}\|_F$
3: $\boldsymbol{Y}_0 \leftarrow \boldsymbol{Y}_0/\alpha, \quad k \leftarrow 0$
4: **while** $k < K$ **do**
5: $\quad \boldsymbol{T}_k \leftarrow 3\boldsymbol{I} - \boldsymbol{Z}_k\boldsymbol{Y}_k$
6: $\quad \boldsymbol{Y}_{k+1} \leftarrow \frac{1}{2}\boldsymbol{Y}_k\boldsymbol{T}_k$
7: $\quad \boldsymbol{Z}_{k+1} \leftarrow \frac{1}{2}\boldsymbol{T}_k\boldsymbol{Z}_k$
8: $\quad k \leftarrow k + 1$
9: **end while**
10: **return** $\sqrt{\alpha}\,\boldsymbol{Y}_K, \; \frac{1}{\sqrt{\alpha}}\,\boldsymbol{Z}_K$

---

A similar matrix-multiplication-only procedure can be used for the quarter-power spectral transformations.

In practice, since

$$\Psi_{1/2}(\boldsymbol{O}_t) = \boldsymbol{O}_t\,(\boldsymbol{O}_t^\top\boldsymbol{O}_t)^{-1/4} = (\boldsymbol{O}_t\boldsymbol{O}_t^\top)^{-1/4}\boldsymbol{O}_t.$$

we can choose $\boldsymbol{X} = \boldsymbol{O}_t^\top\boldsymbol{O}_t \in \mathbb{R}^{n \times n}$ or $\boldsymbol{X} = \boldsymbol{O}_t\boldsymbol{O}_t^\top \in \mathbb{R}^{m \times m}$ depending on whether $m \geq n$ to reduce the computational overhead of the Newton–Schulz iteration. The same applies in the computation of $\Psi_{1/4}$.

**Coupled Newton-Schulz Iteration.** Coupled Newton-Schulz iteration (Higham, 2008) provides an efficient and numerically stable procedure for simultaneously computing the matrix square root $\boldsymbol{X}^{1/2}$ and its inverse $\boldsymbol{X}^{-1/2}$.

Starting from the initialization $\boldsymbol{Y}_0 = \boldsymbol{X}$ and $\boldsymbol{Z}_0 = \boldsymbol{I}$, the method applies a coupled update that repeatedly refines both quantities using only matrix multiplications. Specifically, each iteration is defined as

$$\begin{aligned}
\boldsymbol{Y}_{k+1} &= \frac{1}{2}\boldsymbol{Y}_k\big(3\boldsymbol{I} - \boldsymbol{Z}_k\boldsymbol{Y}_k\big), \\
\boldsymbol{Z}_{k+1} &= \frac{1}{2}\big(3\boldsymbol{I} - \boldsymbol{Z}_k\boldsymbol{Y}_k\big)\boldsymbol{Z}_k,
\end{aligned} \tag{8}$$

which symmetrically updates $\boldsymbol{Y}_k$ and $\boldsymbol{Z}_k$ through the shared correction term $3\boldsymbol{I} - \boldsymbol{Z}_k\boldsymbol{Y}_k$. When appropriately scaled, this coupled iteration converges quadratically, driving $\boldsymbol{Y}_k \to \boldsymbol{X}^{1/2}$ and $\boldsymbol{Z}_k \to \boldsymbol{X}^{-1/2}$ simultaneously.

To ensure numerical stability, a normalization step based on the Frobenius norm of $\boldsymbol{X}$ is applied at initialization and the final iterates are rescaled. In contrast to the standard one-sequence Newton–Schulz iteration, which can be obtained by eliminating one variable from the coupled Newton iteration under a commuting initialization, the coupled formulation simultaneously evaluates $\boldsymbol{Y}_k$ and $\boldsymbol{Z}_k$, making it a more suitable tool for computing matrix roots and inverse roots. Algorithm 1 depicts our method.

# 4. Experiments

## 4.1. Experimental Settings

**Setup.** We conduct our experiments on nanoGPT[1] (Karpathy, 2022), a lightweight GPT-2 training codebase. We follow the standard GPT-2 configuration: GELU activations and a byte-pair encoding tokenizer with vocabulary size 50,257. Our GPT-2 model has 124M parameters. All runs use sequence length 1024, global batch size 480, and are trained on OpenWebText for 200K optimization steps with warmup for 2,000 steps. We do not use QK-Norm or QK-Clip for our primary experiments and conduct all Newton–Schulz iterations with iteration count $K = 5$. Across runs, we vary only the optimizer and its learning rate.

For Muon, we use the reference implementation[2] from Jordan et al. (2024), which applies the matrix update $W_{t+1} = W_t - \eta_t \sqrt{\frac{\text{fan-out}}{\text{fan-in}}} UV^\top$, where $UV^\top$ is the polar factor computed via Algorithm 2.

**Controlled comparison.** To ensure fair and interpretable comparisons, we isolate the optimizer's effect on *matrix-shaped* parameters and, crucially, *decouple matrix and vector learning rates for **all** optimizers*.

Muon is typically evaluated with separate learning rates for matrix and vector parameters, whereas standard Adam baselines often share a single global learning rate. To eliminate this mismatch, we follow Muon and apply Adam to all vector-valued parameters for every method using a fixed learning rate $\text{lr}_{\text{vec}} = 3 \times 10^{-4}$. For matrix parameters, each optimizer is tuned with its own learning rate $\text{lr}_{\text{mat}}$.

We also disable weight decay in the main experiments to isolate the effect of the matrix update rule. Since weight decay enters the update as $W_{t+1} = W_t - \eta_t \Delta W_t - \eta_t \lambda W_t$, jointly tuning $(\eta_t, \lambda)$ would introduce an additional degree of freedom and complicate controlled comparisons. We therefore omit weight decay from the main experiments, and later provide an ablation after fixing the learning rate. We also avoid auxiliary stabilization techniques such as QK-Norm and QK-Clip. All remaining hyperparameters are fixed across methods; in particular, $\beta_1 = 0.9$ and $\beta_2 = 0.95$.

## 4.2. Learning Rate Tuning Protocol

We tune the matrix-parameter learning rate $\text{lr}_{\text{mat}}$ for each optimizer using a two-stage procedure: a coarse logarithmic sweep to identify a stable scale, followed by a local refinement within that scale.

**Coarse search.** We first sweep $\text{lr}_{\text{mat}}$ on a logarithmic grid spanning several orders of magnitude to locate the re-

[1] https://github.com/karpathy/nanoGPT
[2] https://github.com/KellerJordan/Muon

gion where the optimizer transitions from divergence to effective learning. For instance, for AdamW we evaluate $\{10^{-1}, 10^{-2}, 10^{-3}, 10^{-4}, 10^{-5}\}$. This stage intends to identify the correct order of magnitude rather than the exact optimum. Since most candidates in this sweep diverge, we do not plot the resulting curves.

**Fine-grained search.** After identifying a promising scale, we refine the search by evaluating a denser set of learning rates above and below the best coarse candidate. We stop refining once the selected learning rate performs better than its immediate neighbors in this local grid, indicating a stable local optimum within the explored range. For each optimizer, this refinement evaluates roughly ten candidates; for readability, we plot six representative runs in the figures.

## 4.3. Effect of Spectral Exponent on Momentum Inputs

Figure 2 reports representative tuning results when the input $O_t^{\text{mom}}$ is used. We make three observations:

- *Muon stabilizes momentum updates.* Muon (mSGDZ) is considerably more stable than mSGD across a wide learning-rate range. The best learning rate we find for Muon, $7 \times 10^{-3}$, closely matches the value reported in prior benchmarking (Wen et al., 2025).

- *Partial compression improves stability but not to Muon's level.* The intermediate variants mSGDS and mSGDQ are more stable than mSGD, but remain less robust than Muon under aggressive learning rates.

- *After stability is achieved, moderate compression can outperform flattening.* Among the four momentum-input variants, mSGDQ attains the strongest performance once it trains stably, exceeding Muon and obtaining strong results at learning rates of $4 \times 10^{-2}$ and $5 \times 10^{-2}$.

## 4.4. Effect of Spectral Exponent on Normalized Inputs

Similarly, Figure 3 shows our selected results given rms-normalized momentum $O_t^{\text{rms}}$ as input. We observe that:

- *Spectral transforms yield smaller gains when the input is already normalized.* Applying spectral compression to $O_t^{\text{rms}}$ produces limited improvements: AdamS can modestly broaden the stable learning-rate range, whereas stronger compression (AdamQ) offers little benefit and full flattening (AdamZ) degrades performance.

- *All four variants behave similarly at their best settings.* The peak performance differences among Adam, AdamS, and AdamQ are modest, suggesting that elementwise normalization already controls much of the harmful anisotropy that spectral compression targets.

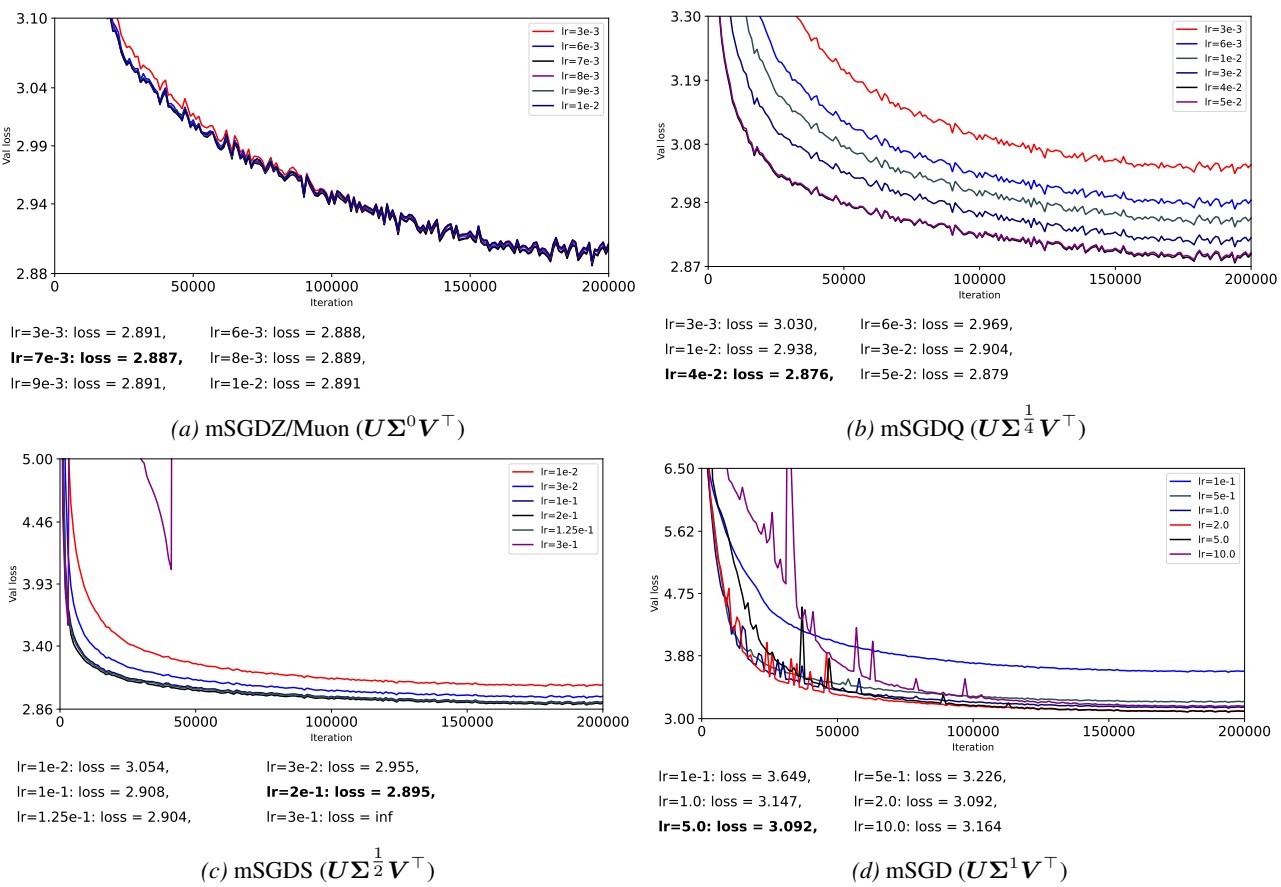

lr=3e-3: loss = 2.891,  lr=6e-3: loss = 2.888,
**lr=7e-3: loss = 2.887,**  lr=8e-3: loss = 2.889,
lr=9e-3: loss = 2.891,  lr=1e-2: loss = 2.891

*(a)* mSGDZ/Muon ($\boldsymbol{U\Sigma^0 V^\top}$)

lr=3e-3: loss = 3.030,  lr=6e-3: loss = 2.969,
lr=1e-2: loss = 2.938,  lr=3e-2: loss = 2.904,
**lr=4e-2: loss = 2.876,**  lr=5e-2: loss = 2.879

*(b)* mSGDQ ($\boldsymbol{U\Sigma^{\frac{1}{4}} V^\top}$)

lr=1e-2: loss = 3.054,  lr=3e-2: loss = 2.955,
lr=1e-1: loss = 2.908,  **lr=2e-1: loss = 2.895,**
lr=1.25e-1: loss = 2.904,  lr=3e-1: loss = inf

*(c)* mSGDS ($\boldsymbol{U\Sigma^{\frac{1}{2}} V^\top}$)

lr=1e-1: loss = 3.649,  lr=5e-1: loss = 3.226,
lr=1.0: loss = 3.147,  lr=2.0: loss = 3.092,
**lr=5.0: loss = 3.092,**  lr=10.0: loss = 3.164

*(d)* mSGD ($\boldsymbol{U\Sigma^1 V^\top}$)

*Figure 2.* Overall comparison across four optimizers (mSGDZ/Muon, mSGDQ, mSGDS and mSGD) based on the **first-moment momentum $\boldsymbol{M}_t$**. Each subfigure corresponds to a different optimizer.

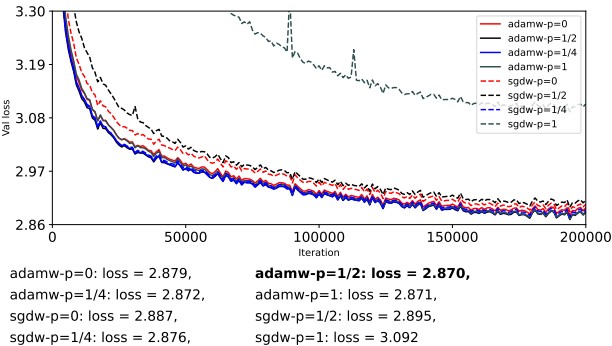

adamw-p=0: loss = 2.879,  **adamw-p=1/2: loss = 2.870,**
adamw-p=1/4: loss = 2.872,  adamw-p=1: loss = 2.871,
sgdw-p=0: loss = 2.887,  sgdw-p=1/2: loss = 2.895,
sgdw-p=1/4: loss = 2.876,  sgdw-p=1: loss = 3.092

*Figure 4.* Comparison of four spectral exponents $p \in \{0, \frac{1}{4}, \frac{1}{2}, 1\}$ applied to either the first-moment momentum $\boldsymbol{M}_t$ or the RMS-normalized update $\boldsymbol{M}_t \oslash \sqrt{\boldsymbol{V}_t}$, each shown at its tuned learning rate. Dashed curves correspond to momentum-input variants, while solid curves correspond to RMS-normalized-input variants.

### 4.5. Momentum vs. RMS-Normalized Inputs

In Figure 4, we compare the best-tuned run from each of the eight configurations. Two trends are clear. First, switching the input from the momentum $\boldsymbol{O}_t^{\text{mom}}$ to the RMS-normalized

update $\boldsymbol{O}_t^{\text{rms}}$ consistently improves both stability and final performance across spectral exponents. In particular, AdamZ (Muon-style orthogonalization applied to $\boldsymbol{O}_t^{\text{rms}}$) is stronger than mSGDZ (the original Muon applied to $\boldsymbol{O}_t^{\text{mom}}$). Second, within the RMS-normalized family, performance differences across $p$ are relatively small, with AdamS emerging as a strong and stable choice.

### 4.6. Additional Ablations and Comparisons

**Comparison under weight decay and QK-Norm.** Our main experiments disable weight decay and auxiliary stabilization techniques to isolate the effect of the update rule. To determine whether our conclusions persist under more practical settings, we additionally evaluate the RMS-normalized spectral variants with weight decay and QK-Norm enabled. The full training curves are provided in Appendix E.

Since weight decay enters the parameter update through the coupled product $\eta\lambda$, we start from the best learning rate selected for each Adam variant in Figure 3, and choose the corresponding $\lambda$ such that $\eta\lambda$ matches the Adam baseline value $4 \times 10^{-3} \cdot 0.1$.

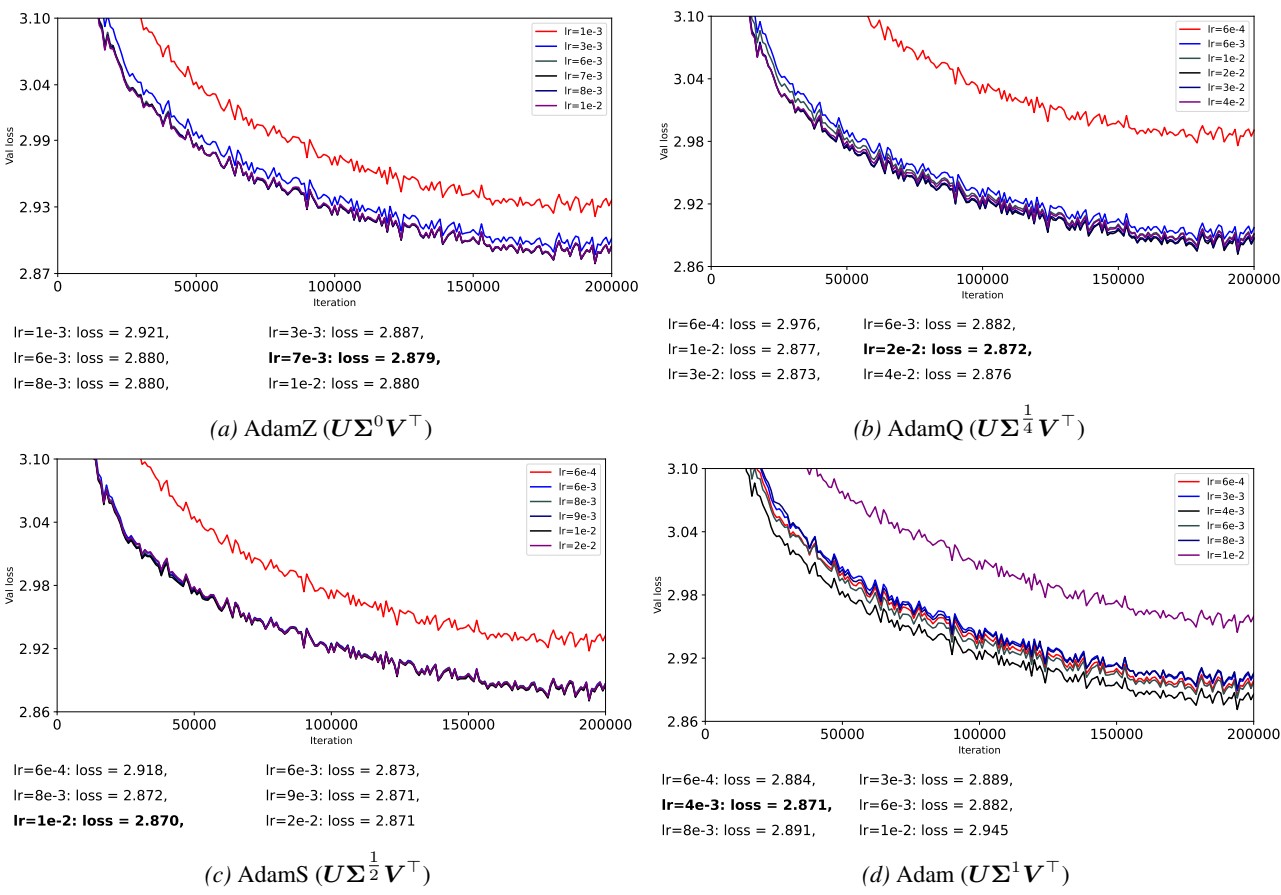

lr=1e-3: loss = 2.921,     lr=3e-3: loss = 2.887,
lr=6e-3: loss = 2.880,     **lr=7e-3: loss = 2.879,**
lr=8e-3: loss = 2.880,     lr=1e-2: loss = 2.880

*(a)* AdamZ $(U\Sigma^0 V^\top)$

lr=6e-4: loss = 2.976,     lr=6e-3: loss = 2.882,
lr=1e-2: loss = 2.877,     **lr=2e-2: loss = 2.872,**
lr=3e-2: loss = 2.873,     lr=4e-2: loss = 2.876

*(b)* AdamQ $(U\Sigma^{\frac{1}{4}} V^\top)$

lr=6e-4: loss = 2.918,     lr=6e-3: loss = 2.873,
lr=8e-3: loss = 2.872,     lr=9e-3: loss = 2.871,
**lr=1e-2: loss = 2.870,**     lr=2e-2: loss = 2.871

*(c)* AdamS $(U\Sigma^{\frac{1}{2}} V^\top)$

lr=6e-4: loss = 2.884,     lr=3e-3: loss = 2.889,
**lr=4e-3: loss = 2.871,**     lr=6e-3: loss = 2.882,
lr=8e-3: loss = 2.891,     lr=1e-2: loss = 2.945

*(d)* Adam $(U\Sigma^1 V^\top)$

*Figure 3.* Overall comparison across four optimizers (AdamZ, AdamS, AdamQ and Adam) based on the **second-moment-normalized update** $M_t \oslash \sqrt{V_t}$. Each subfigure corresponds to a different optimizer.

With both weight decay and QK-Norm, AdamZ, AdamS, AdamQ, and Adam reach final validation losses of 2.806, 2.802, 2.800, and 2.815, respectively. Weight decay improves all RMS-normalized variants, and QK-Norm provides a further gain. Importantly, the relative trend remains consistent with our main finding: spectral compression offers useful stabilization and performance benefits, but full spectral flattening is not uniformly superior, especially once RMS normalization and other stabilizers are incorporated.

**Effect of Newton–Schulz iteration count.** We also study the sensitivity of the spectral transform to the number of Newton–Schulz iterations in Appendix F. Even with synthetic inputs that span a large range of singular values, the coupled NS iteration algorithm with $K = 5$ yields results with high cosine similarity to those from direct SVD.

Moreover, on our experimental setup, increasing the iteration count from $K = 5$ to $K = 15$ produces nearly identical training curves and final validation losses. This suggests that $K = 5$ is sufficient for the optimizer comparisons reported in the main experiments.

### 4.7. Runtime Overhead of Newton–Schulz Iterations

Newton–Schulz iterations introduce only moderate training overhead in our setting. For $W \in \mathbb{R}^{m \times n}$ with $n \leq m$, a $K$-step NS transform costs $O(Kn^3 + mn^2)$, where $Kn^3$ comes from repeated operations on the smaller $n \times n$ Gram matrix, and $mn^2$ comes from forming the Gram matrix and multiplying the spectral factor back to the original update.

Importantly, this cost is independent of the batch size per step, so its relative contribution to the overall training cost decreases as the batch size increases. The main additional memory overhead comes from transient $n \times n$ matrices. With $K = 5$ on an 8-GPU A800 node, the wall-clock overhead across Adam spectral variants is shown in Table 1.

*Table 1.* Runtime overhead on an 8-GPU A800 node.

|  | Adam | AdamS | AdamQ | AdamZ |
|---|---|---|---|---|
| Runtime | 33h 05m | 34h 18m | 34h 21m | 34h 09m |
| Overhead | – | 3.7% | 3.8% | 3.2% |

# 5. Discussion: Our Understanding of Muon

The primary goal of this work is to evaluate when Muon-style orthogonalization (and, more broadly, spectral compression) is a reliable optimization strategy. Accordingly, we interpret Muon's empirical behavior in Section 4 through three honest conclusions.

**I. Muon exhibits no significant performance edge over AdamW in our experimental regime.** In our controlled setting, Muon-style orthogonalization does not consistently outperform Adam, whether applied to first-moment momentum or to RMS-normalized updates. This contrasts with prior reports of Muon converging faster than Adam (Jordan et al., 2024). The discrepancy is likely due to differences in methodology, which include learning-rate decoupling, omission of auxiliary stabilizers, disabled weight decay, and a relatively small 124M-parameter model scale.

**II. Muon is stable but aggressive.** On first-moment momentum inputs, Muon provides a clear stabilization effect where orthogonalization significantly improves robustness over mSGD across most learning rates (Figure 2). This behavior aligns with Muon's defining operation: given $G_t = U\Sigma V^\top$, Muon replaces the spectrum by $\Sigma^0$, producing the polar factor $UV^\top$. By discarding singular-value magnitudes, Muon prevents directions with extremely large singular values from dominating the update, mitigating ill-conditioning through *direction-wise* spectral scale control.

However, this stabilization mechanism is also indiscriminate. Muon applies the same singular-value scaling to every direction, independent of curvature or signal strength. While momentum scale is not necessarily an optimal step-size signal, full spectral flattening removes all magnitude distinctions among singular directions. Thus, when some directions are legitimately more informative than others, flattening them to unity can reduce their relative influence. In this sense, Muon stabilizes by construction, but it does not selectively preserve meaningful anisotropy when such anisotropy may be beneficial.

**III. Muon may magnify noisy gradient directions.** Muon's biggest limitation also follows from its complete spectral flattening: by mapping all nonzero singular values to 1, it removes magnitude-based filtering. Directions corresponding to small singular values are no longer suppressed relative to dominant directions and can receive comparable update weight. This is especially problematic in the later stages of training where update matrices typically have low effective rank, and small-singular-value components may be primarily dominated by noise.

This effect is consistent with our observation that aggressive Muon-style orthogonalization on RMS-normalized inputs does not consistently outperform Adam and is often worse than partial compression, which can offer a more balanced trade-off between stabilization and preserving original spectral structure.

**Overall.** Together, our results suggest that spectral compression can provide genuine stabilization benefits, especially when the input update is *not* already normalized. However, full orthogonalization ($p = 0$) is not always desirable: its indiscriminate flattening can suppress informative dominant modes while amplifying low-signal modes. Hence, our experiments do not support orthogonalization as a universally superior replacement for modern RMS-normalized second-moment optimizers.

# 6. Conclusion and Limitations

**Conclusion.** This paper studies Muon and related matrix-based optimizers from a unified spectral perspective. By viewing Muon as the $p = 0$ endpoint of the family $\Psi_p(O) = U\Sigma^p V^\top$, we separate the effect of spectral compression from other optimizer design choices. Our controlled experiments show that spectral compression provides a clear stabilization benefit for first-moment updates, but full Muon-style flattening is not uniformly superior once RMS-normalized second-moment updates are used. These results suggest that Muon is best understood as an effective spectral normalization mechanism rather than a universally superior replacement for Adam-style optimizers.

**Limitations and future work.** Our study is intentionally conducted in a controlled nanoGPT/OpenWebText regime to isolate the effect of spectral transformations from confounding factors such as weight decay, auxiliary normalization, and optimizer-specific tuning. While this design makes the comparison more interpretable, it also limits the scope of our empirical conclusions. In particular, we do not claim that the relative ranking of optimizers will necessarily hold across larger model scales, different architectures, longer training runs, or fully tuned production training recipes.

Although our ablations include more practical ingredients such as weight decay and QK-Norm, which are omitted from the main experiments, these studies are still limited in scope. A more complete investigation of how spectral optimizers interact with weight decay, QK-Norm, learning-rate decay, and other stabilization techniques is an important direction for future work.

Finally, fractional spectral variants rely on coupled Newton–Schulz iterations, which introduce moderate runtime and memory overhead. Improving the efficiency of these matrix-function approximations is therefore another important direction for making spectral compression more practical at larger scales.

## Acknowledgment

This work is partly supported by the National Natural Science Foundation of China under Grant No. 62576048.

## Impact Statement

"This paper presents work whose goal is to advance the field of Machine Learning. There are many potential societal consequences of our work, none which we feel must be specifically highlighted here."

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

# A. Derivation of $UV^\top$ based on Newton-Schulz method

Let the singular value decomposition of a matrix $M \in \mathbb{R}^{m \times n}$ be

$$M = USV^\top,$$

where $U \in \mathbb{R}^{m \times r}, V \in \mathbb{R}^{n \times r}, S = \mathrm{Diag}(\sigma_1, \ldots, \sigma_r), \sigma_i > 0$, and $U, V$ have orthonormal columns. Let us assume $m \leq n$

Our goal is to compute the orthogonal (polar) factor $UV^\top$ *without explicitly performing an SVD*, using the Newton–Schulz iteration.

**Step 1: Express $UV^\top$ as a matrix function of $M$.** We start from quantities that can be formed directly from $M$. Consider $M^\top M \in \mathbb{R}^{n \times n}$. Substituting the SVD of $M$ gives

$$M^\top M = (USV^\top)^\top (USV^\top) = VSU^\top USV^\top = VS^2V^\top,$$

where we used $U^\top U = I_r$.

Hence, the inverse square root of $M^\top M$ (on its rank-$r$ subspace) is

$$(M^\top M)^{-1/2} = VS^{-1}V^\top.$$

Left-multiplying by $M$ yields

$$M(M^\top M)^{-1/2} = (USV^\top)(VS^{-1}V^\top) = U(SS^{-1})V^\top = UV^\top.$$

This identity shows that $UV^\top$ can be obtained by removing the singular-value scaling of $M$ via $(M^\top M)^{-1/2}$.

**Step 2: Reduction to an inverse square root problem.** Define

$$A := M^\top M \succeq 0.$$

The problem is now reduced to computing $A^{-1/2}$ using only matrix multiplication, without SVD.

**Step 3: Newton–Schulz iteration for the inverse square root.** Let us consider a scalar $a > 0$, consider the problem of computing $z = a^{-1/2}$. This is equivalent to solving $f(z) = \frac{1}{z^2} - a = 0$. Applying Newton's method gives

$$z^+ = z - \frac{f(z)}{f'(z)} = z - \frac{z^{-2} - a}{-2z^{-3}} = \frac{1}{2} z(3 - az^2).$$

This is the scalar Newton–Schulz update.

Replacing the scalar $a$ by a matrix $A \succeq 0$, the scalar $z$ by a matrix $Z$, and scalar multiplication by matrix multiplication yields the matrix Newton–Schulz iteration

$$Z_{k+1} = \frac{1}{2} Z_k (3I - AZ_k^2).$$

This iteration is equal to Equation 4 in the main body. It involves only matrix multiplication, addition, and scalar scaling. After sufficient iterations,

$$Z_k \approx A^{-1/2}.$$

**Step 4: Recovering $UV^\top$.** Using the identity $UV^\top = MA^{-1/2}$, $A = M^\top M$, and the approximation $Z_k \approx A^{-1/2}$, we obtain

$$UV^\top \approx MZ_k.$$

## B. Newton-Schulz method used by Jordan et al. (2024)

---
**Algorithm 2** Newton-Schulz to solve $UV^\top$ where $M = U\Sigma V^\top$

---
**Require:** $M, K$
**Ensure:** $UV^\top$
 1: **Coefficients:** $(a, b, c) \leftarrow (3.4445, -4.7750, 2.0315)$
 2: $Z_0 \leftarrow M$
 3: $\alpha \leftarrow \|Z_0\|_F$
 4: $Z_0 \leftarrow Z_0/\alpha,\ k \leftarrow 0$
 5: **while** $k < K$ **do**
 6: $\quad Z_{k+1} \leftarrow a\,Z_k + b\,Z_k Z_k^\top Z_k + c\,(Z_k Z_k^\top)^2 Z_k$
 7: $\quad k \leftarrow k + 1$
 8: **end while**
 9: **return** $Z_K$

---

## C. Derive $X^{\frac{1}{2}}$ based on Coupled Newton-Schulz method

We aim to compute the matrix square root $X^{1/2}$. Let the unknown matrix $Y$ satisfy

$$f(Y) := Y^2 - X = 0.$$

This defines a matrix equation $f : \mathbb{C}^{n \times n} \longrightarrow \mathbb{C}^{n \times n}$.

**Fréchet Derivative.** For an arbitrary perturbation $H$, we have

$$f(Y + H) = (Y + H)^2 - X = Y^2 - X + (YH + HY) + H^2.$$

Therefore, the Fréchet derivative of $f$ at $Y$ is

$$Df(Y)[H] = YH + HY.$$

**Definition of the Newton Step.** The Newton increment $\Delta Y$ at $Y$ is defined as the solution of

$$Df(Y)[\Delta Y] = -f(Y),$$

that is, the Sylvester equation

$$Y\Delta Y + \Delta Y\,Y = -(Y^2 - X). \tag{9}$$

The update is then given by

$$Y^+ = Y + \Delta Y.$$

Up to this point, the derivation corresponds to the *exact matrix Newton method*, without any approximation.

**A Coupled Newton System.** The main difficulty of (9) is that it requires solving a Sylvester equation. A classical technique is to couple the unknown square root with its inverse, thereby converting the Sylvester solve into matrix multiplications.

Let $Z$ approximate $Y^{-1}$ (and ultimately $X^{-1/2}$). Consider the coupled system

$$\begin{cases} F_1(Y, Z) := Y^2 - X = 0, \\ F_2(Y, Z) := ZY - I = 0. \end{cases} \tag{10}$$

The exact solution is

$$(Y_\star, Z_\star) = (X^{1/2}, X^{-1/2}).$$

**Fréchet Derivative of the Coupled System** For perturbations $(\Delta Y, \Delta Z)$:

- For $F_1$,
$$DF_1(Y, Z)[\Delta Y, \Delta Z] = Y\Delta Y + \Delta Y\, Y.$$

- For $F_2$,
$$(Z + \Delta Z)(Y + \Delta Y) - I = (ZY - I) + (Z\Delta Y + \Delta Z\, Y) + (\Delta Z)(\Delta Y),$$

  hence
$$DF_2(Y, Z)[\Delta Y, \Delta Z] = Z\Delta Y + \Delta Z\, Y.$$

**Newton Linear System (Exact)** The Newton step $(\Delta Y, \Delta Z)$ is defined by

$$DF(Y, Z)[\Delta Y, \Delta Z] = -F(Y, Z),$$

that is,

$$\begin{aligned} Y\Delta Y + \Delta Y\, Y &= -(Y^2 - X), \\ Z\Delta Y + \Delta Z\, Y &= -(ZY - I). \end{aligned} \tag{11}$$

This system is still an *exact Newton system*.

**A Closed-Form Newton Step via a Symmetric Multiplicative Ansatz** We adopt the symmetric multiplicative ansatz

$$Y^+ = YQ, \qquad Z^+ = QZ, \tag{12}$$

where $Q = Q(Y, Z)$ is to be determined.

Define

$$R := ZY.$$

Then

$$R^+ := Z^+Y^+ = (QZ)(YQ) = Q(ZY)Q = QRQ.$$

Hence, choosing $Q$ reduces to the problem: given $R$, construct $Q$ such that $QRQ \approx I$.

**Deriving $Q = \frac{1}{2}(3I - ZY)$ from Newton's Principle.** To approximate $R^{-1/2}$, ideally,

$$Q = R^{-1/2} \quad \Rightarrow \quad QRQ = I.$$

Consider the matrix function

$$\phi(R) := R^{-1/2}.$$

When $R$ is close to the identity, write

$$R = I + E, \qquad \|E\| \ll 1.$$

Accoording to the Taylor expansion, we have

$$(I + E)^{-1/2} = I - \tfrac{1}{2}E + O(\|E\|^2).$$

Substituting $E = R - I$, we obtain

$$R^{-1/2} \approx I - \tfrac{1}{2}(R - I) = \tfrac{1}{2}(3I - R).$$

Thus define

$$Q := \tfrac{1}{2}(3I - R) = \tfrac{1}{2}(3I - ZY). \tag{13}$$

**Deriving $Q = \tfrac{1}{2}(3I - ZY)$ from Newton's Principle**   Coupled Newton–Schulz Iteration.

Substituting (13) into (12), we obtain

$$Y^+ = \tfrac{1}{2}Y(3I - ZY),$$
$$Z^+ = \tfrac{1}{2}(3I - ZY)Z.$$

Write it as an iteration equation, we have

$$
\begin{aligned}
Y_{k+1} &= \tfrac{1}{2}Y_k(3I - Z_kY_k), & Y_0 &= X, \\
Z_{k+1} &= \tfrac{1}{2}(3I - Z_kY_k)Z_k, & Z_0 &= I.
\end{aligned}
\tag{14}
$$

This is the *coupled Newton–Schulz iteration*, which simultaneously drives

$$Y_k \to X^{1/2}, \qquad Z_k \to X^{-1/2}.$$

# D. Derive $X^{\frac{1}{4}}$ based on Coupled Newton-Schulz method

---

**Algorithm 3** Two-times Coupled Newton-Schulz to solve $X^{\frac{1}{4}}$ and $X^{\frac{-1}{4}}$

---

**Require:** $X, K$
**Ensure:** $X^{\frac{1}{4}}, X^{\frac{-1}{4}}$
1: $Y_0 \leftarrow X, Z_0 \leftarrow I$
2: $\alpha \leftarrow \|X\|_F$
3: $Y_0 \leftarrow \frac{Y_0}{\alpha}, k \leftarrow 0$
4: **while** $k < K$ **do**
5:     $T_k \leftarrow 3I - Z_kY_k$
6:     $Y_{k+1} \leftarrow \tfrac{1}{2}Y_kT_k$
7:     $Z_{k+1} \leftarrow \tfrac{1}{2}T_kZ_k$
8:     $k \leftarrow k + 1$
9: **end while**
10: $Y_0 \leftarrow \sqrt{\alpha}Y_K, Z_0 \leftarrow I$
11: $\beta \leftarrow \|Y_0\|_F$
12: $Y_0 \leftarrow \frac{Y_0}{\beta}, k \leftarrow 0$
13: **while** $k < K$ **do**
14:     $T_k \leftarrow 3I - Z_kY_k$
15:     $Y_{k+1} \leftarrow \tfrac{1}{2}Y_kT_k$
16:     $Z_{k+1} \leftarrow \tfrac{1}{2}T_kZ_k$
17:     $k \leftarrow k + 1$
18: **end while**
19: **return** $\sqrt{\beta}Y_K, \frac{1}{\sqrt{\beta}}Z_K$

---

# E. Additional Comparisons under Weight Decay and QK-Norm

We provide the full training curves for the additional comparisons with weight decay and QK-Norm in Figure 5.

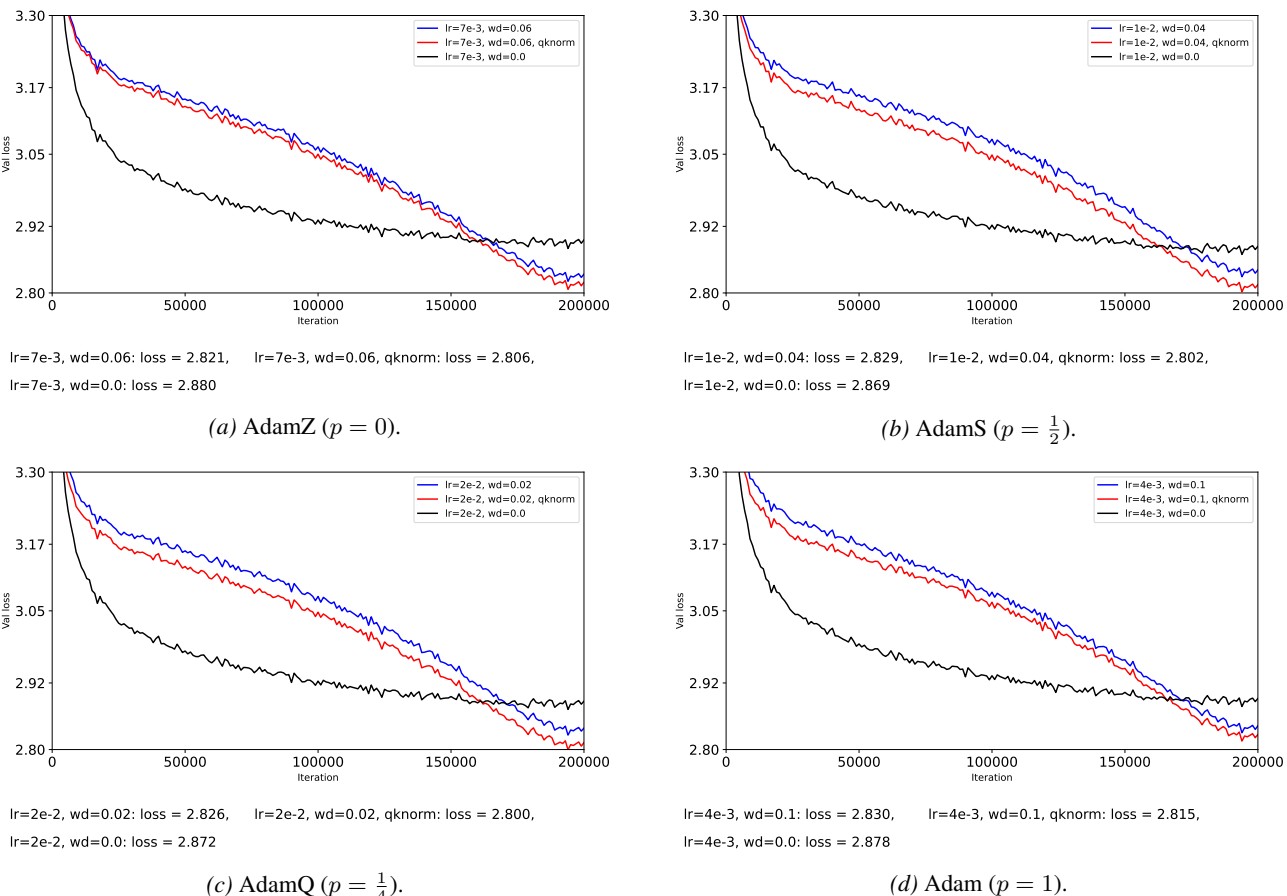

lr=7e-3, wd=0.06: loss = 2.821,    lr=7e-3, wd=0.06, qknorm: loss = 2.806,
lr=7e-3, wd=0.0: loss = 2.880

*(a)* AdamZ ($p = 0$).

lr=1e-2, wd=0.04: loss = 2.829,    lr=1e-2, wd=0.04, qknorm: loss = 2.802,
lr=1e-2, wd=0.0: loss = 2.869

*(b)* AdamS ($p = \frac{1}{2}$).

lr=2e-2, wd=0.02: loss = 2.826,    lr=2e-2, wd=0.02, qknorm: loss = 2.800,
lr=2e-2, wd=0.0: loss = 2.872

*(c)* AdamQ ($p = \frac{1}{4}$).

lr=4e-3, wd=0.1: loss = 2.830,    lr=4e-3, wd=0.1, qknorm: loss = 2.815,
lr=4e-3, wd=0.0: loss = 2.878

*(d)* Adam ($p = 1$).

*Figure 5.* Additional comparisons under weight decay and QK-Norm across RMS-normalized spectral variants. Each panel compares training without weight decay, with weight decay, and with both weight decay and QK-Norm.

# F. Ablation of Newton–Schulz Iteration Count

To evaluate the approximation quality of the coupled Newton–Schulz (NS) procedure, we compare the NS approximation against the exact SVD-based spectral transform. Specifically, we generate Gaussian matrices of sizes $512 \times 512$ and $1024 \times 1024$ with condition numbers ranging from 1 to 100. For each matrix $O$, we compute the exact update $\Psi_p(O)$ using SVD and compare it with the $K$-step NS approximation using cosine similarity between the vectorized update matrices. Representative results are shown in Table 2.

*Table 2.* Cosine similarity between NS approximation and the exact SVD-based spectral transform for synthetic inputs.

| $K$ | cos-sim ($p = \frac{1}{2}$) | cos-sim ($p = \frac{1}{4}$) |
|---|---|---|
| 3 | 0.990 | 0.975 |
| 5 | 0.997 | 0.993 |
| 7 | 0.999 | 0.999 |
| 10 | 1.000 | 1.000 |

The approximation improves rapidly as $K$ increases. By $K = 5$, both $p = \frac{1}{2}$ and $p = \frac{1}{4}$ already achieve very high cosine similarity with the exact SVD solution. Therefore, we use $K = 5$ in our main experiments.

We also verify whether increasing the number of NS iterations affects actual training behavior. In Figure 6, we compare AdamS ($p = \frac{1}{2}$) with $K = 5$ and $K = 15$ under the same setting. The two training curves are nearly identical, and the final validation losses are also very close.

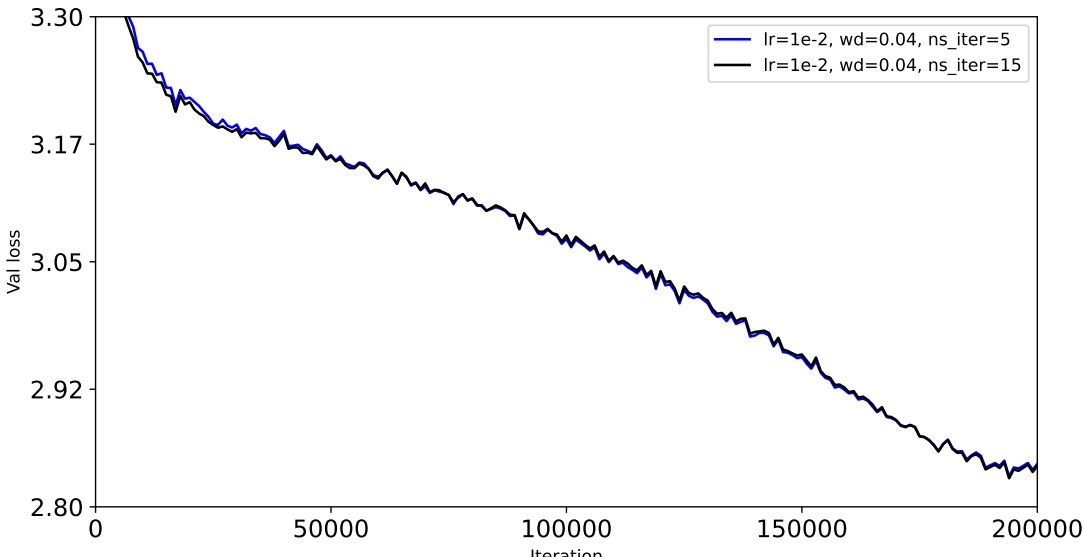

lr=1e-2, wd=0.04, ns_iter=5: loss = 2.831,

lr=1e-2, wd=0.04, ns_iter=15: loss = 2.829

*Figure 6.* Effect of Newton–Schulz iteration count on training for AdamS ($p = \frac{1}{2}$) with $K = 5$ and $K = 15$ NS iterations.

# G. Detailed results for different settings

*Table 3.* **Hyperparameter sweeps for Muon.**

| Origin | $\Psi_p(\boldsymbol{O}^{\mathbf{mom}})$ | Learning Rate | Weight Decay | Val Loss |
|---|---|---|---|---|
| First order | $\boldsymbol{U\Sigma^1 V}^\top$ | 1e-1 | 0 | 3.649 |
| First order | $\boldsymbol{U\Sigma^1 V}^\top$ | 5e-1 | 0 | 3.226 |
| First order | $\boldsymbol{U\Sigma^1 V}^\top$ | 1.0 | 0 | 3.147 |
| First order | $\boldsymbol{U\Sigma^1 V}^\top$ | 2.0 | 0 | **3.092** |
| First order | $\boldsymbol{U\Sigma^1 V}^\top$ | 5.0 | 0 | 3.092 |
| First order | $\boldsymbol{U\Sigma^1 V}^\top$ | 10.0 | 0 | 3.164 |
| First order | $\boldsymbol{U\Sigma^{\frac{1}{2}} V}^\top$ | 1e-2 | 0 | 3.054 |
| First order | $\boldsymbol{U\Sigma^{\frac{1}{2}} V}^\top$ | 3e-2 | 0 | 2.955 |
| First order | $\boldsymbol{U\Sigma^{\frac{1}{2}} V}^\top$ | 1e-1 | 0 | 2.908 |
| First order | $\boldsymbol{U\Sigma^{\frac{1}{2}} V}^\top$ | 2e-1 | 0 | **2.895** |
| First order | $\boldsymbol{U\Sigma^{\frac{1}{2}} V}^\top$ | 1.25e-1 | 0 | 2.904 |
| First order | $\boldsymbol{U\Sigma^{\frac{1}{2}} V}^\top$ | 3e-1 | 0 | inf |
| First order | $\boldsymbol{U\Sigma^{\frac{1}{4}} V}^\top$ | 3e-3 | 0 | 3.030 |
| First order | $\boldsymbol{U\Sigma^{\frac{1}{4}} V}^\top$ | 6e-3 | 0 | 2.969 |
| First order | $\boldsymbol{U\Sigma^{\frac{1}{4}} V}^\top$ | 1e-2 | 0 | 2.938 |
| First order | $\boldsymbol{U\Sigma^{\frac{1}{4}} V}^\top$ | 3e-2 | 0 | 2.904 |
| First order | $\boldsymbol{U\Sigma^{\frac{1}{4}} V}^\top$ | 4e-2 | 0 | **2.876** |
| First order | $\boldsymbol{U\Sigma^{\frac{1}{4}} V}^\top$ | 5e-2 | 0 | 2.879 |
| First order | $\boldsymbol{U\Sigma^0 V}^\top$ | 3e-3 | 0 | 2.891 |
| First order | $\boldsymbol{U\Sigma^0 V}^\top$ | 6e-3 | 0 | 2.888 |
| First order | $\boldsymbol{U\Sigma^0 V}^\top$ | 7e-3 | 0 | **2.887** |
| First order | $\boldsymbol{U\Sigma^0 V}^\top$ | 8e-3 | 0 | 2.889 |
| First order | $\boldsymbol{U\Sigma^0 V}^\top$ | 9e-3 | 0 | 2.891 |
| First order | $\boldsymbol{U\Sigma^0 V}^\top$ | 1e-2 | 0 | 2.891 |

*Table 4.* **Hyperparameter sweeps for Muon.**

| Origin | $\Psi_p(O^{\mathbf{rms}})$ | Learning Rate | Weight Decay | Val Loss |
|---|---|---|---|---|
| Second order | $U\Sigma^1 V^\top$ | 6e-4 | 0 | 2.884 |
| Second order | $U\Sigma^1 V^\top$ | 3e-3 | 0 | 2.889 |
| Second order | $U\Sigma^1 V^\top$ | 4e-3 | 0 | **2.871** |
| Second order | $U\Sigma^1 V^\top$ | 6e-3 | 0 | 2.882 |
| Second order | $U\Sigma^1 V^\top$ | 8e-3 | 0 | 2.891 |
| Second order | $U\Sigma^1 V^\top$ | 1e-2 | 0 | 2.945 |
| Second order | $U\Sigma^{\frac{1}{2}} V^\top$ | 6e-4 | 0 | 2.918 |
| Second order | $U\Sigma^{\frac{1}{2}} V^\top$ | 6e-3 | 0 | 2.873 |
| Second order | $U\Sigma^{\frac{1}{2}} V^\top$ | 8e-3 | 0 | 2.872 |
| Second order | $U\Sigma^{\frac{1}{2}} V^\top$ | 9e-3 | 0 | 2.871 |
| Second order | $U\Sigma^{\frac{1}{2}} V^\top$ | 1e-2 | 0 | **2.870** |
| Second order | $U\Sigma^{\frac{1}{2}} V^\top$ | 2e-2 | 0 | 2.871 |
| Second order | $U\Sigma^{\frac{1}{4}} V^\top$ | 6e-4 | 0 | 2.976 |
| Second order | $U\Sigma^{\frac{1}{4}} V^\top$ | 6e-3 | 0 | 2.882 |
| Second order | $U\Sigma^{\frac{1}{4}} V^\top$ | 1e-2 | 0 | 2.877 |
| Second order | $U\Sigma^{\frac{1}{4}} V^\top$ | 2e-2 | 0 | **2.872** |
| Second order | $U\Sigma^{\frac{1}{4}} V^\top$ | 3e-2 | 0 | 2.873 |
| Second order | $U\Sigma^{\frac{1}{4}} V^\top$ | 4e-2 | 0 | 2.876 |
| Second order | $U\Sigma^0 V^\top$ | 1e-3 | 0 | 2.921 |
| Second order | $U\Sigma^0 V^\top$ | 3e-3 | 0 | 2.887 |
| Second order | $U\Sigma^0 V^\top$ | 6e-3 | 0 | 2.880 |
| Second order | $U\Sigma^0 V^\top$ | 7e-3 | 0 | **2.879** |
| Second order | $U\Sigma^0 V^\top$ | 8e-3 | 0 | 2.880 |
| Second order | $U\Sigma^0 V^\top$ | 1e-2 | 0 | 2.880 |

