# OpenReview forum: "Delving into Muon and Beyond: Deep Analysis and Extensions"
_ICML.cc/2026/Conference — ICML 2026 spotlight_

### Official Review · Reviewer_k39R · 2026-02-17

**Soundness:** 3
**Presentation:** 4
**Significance:** 3
**Originality:** 4
**Overall Recommendation:** 5
**Confidence:** 4

**Summary:**

The paper's main contribution consists of reframing Muon as one instance in a broader spectral compression continuum, interpolating between Muon and the original spectrum, and providing an empirical assessment of the algorithms within this family under tightly controlled settings. It studies optimizers induced by spectral transformations of the form $U\Sigma^pV^T$, focusing on $p\in\{0,1/4,1/2,1\}$. In this view, Muon corresponds to the endpoint $p=0$, which replaces the singular values by 1, while $p=1$ recovers the identity transformation. Intermediate exponents $p=1/2$ and $p=1/4$ interpolate between identity and full spectral flattening.
To make fractional spectral transformations practical, the paper develops a coupled Newton–Schulz iteration to compute matrix power transformations without explicit SVD. The authors then conduct controlled experiments on NanoGPT with OpenWebText, decoupling matrix and vector learning rates, disabling weight decay, and avoiding auxiliary techniques (e.g., QK-Norm/Clip), in order to isolate the effect of spectral transformations.
Empirically, they find: (1) Muon ($p=0$) stabilizes momentum updates (2) when applied to RMS-normalized updates, full orthogonalization does not outperform AdamW and may underperform partial spectral compression (3) orthogonalization is not universally superior to RMS-normalized second-moment optimizers.

**Compliance With Llm Reviewing Policy:**

Affirmed.

**Final Justification:**

I maintain my view that the paper is solid and keep my score unchanged.

**Key Questions For Authors:**

1. Why is $\Psi_p$ applied to $O^{rms}_t$ rather than applying the transformations in the reverse order?

2. Given the observations in the paper, have the authors considered adapting $p$ during training or using different values of $p$ for different layers? If so, do they have insights into how the appropriate transformation might depend on layer type?

**Limitations:**

Yes

**Strengths And Weaknesses:**

**Soundness**

Strengths:
- The submission is technically sound and the empirical claims are well supported by experimental results.
- Experimental design is well-controlled: decoupled matrix/vector learning rates, no weight decay, no auxiliary stabilizers. The tuning protocol is systematic, clearly described, and enables a fair comparison across methods.
- The authors provide an honest interpretation of the results and avoid exaggerated claims.
- Limitations are explicitly discussed.

Weaknesses:
- Experiments are limited to a single model size and a single training setup.

**Presentation**

- The paper is clearly written, well structured and easy to follow.
- Discussion section is honest about limitations.
- Good literature overview and positioning relative to prior matrix-based optimizers and Adam-style methods.

**Significance**

- Given the growing popularity of Muon, the paper addresses a timely and relevant question whether Muon is fundamentally superior to Adam-style second-moment methods. The conclusion that this need not be the case is interesting and potentially impactful.
- The paper provides conceptual clarification rather than a new SOTA optimizer, but its conclusions may influence future work on matrix-based optimizers and their design.

**Originality**

- Very simple yet interesting idea that to my knowledge has not been considered before.
- Novel unification of Muon within a continuous spectral family applying updates of the form $U\Sigma^pV^T$.
- New framing of partial spectrum compression as controlled anisotropy reduction.
- Coupled Newton–Schulz is technically incremental but well integrated.

---

> ### Author Rebuttal · Authors · 2026-03-31
>
> We sincerely thank the reviewer for appreciating this paper. We are encouraged that the reviewer found the paper technically sound, the spectral framing interesting, and the conclusions about Muon useful. _In this paper, we aim to provide a simple, practical and honest analysis of Adam and Muon in Transformer model._
>
> Below, we address your main concerns on experimental scope, transformation order, and adaptive choices of $p$. In the rebuttal stage, we supply more ablation studies.
>
> We really hope our responses resolve your concerns.
>
> ---
>
> **Q1: Limited Experimental Scope**
> > Experiments are limited to a single model size and a single training setup.
>
> Response: Thank you very much for pointing this out. We agree that evaluating a broader range of model sizes and training setups would further strengthen the paper.
>
> That said, our current setup is a standard and representative nanoGPT-style training regime. Specifically, all models are trained on 100B tokens, with each iteration processing roughly 500K tokens. This scale and configuration are consistent with widely used nanoGPT-based experimental settings and with several recent studies in the same line of work [1,2].
>
> Our goal in this paper was to isolate the effect of spectral compression in a tightly controlled setting. Doing so required substantial optimizer-specific tuning, which made it important to first establish the phenomenon clearly in one stable and well-understood regime before expanding to additional scales and architectures. We will clarify this scope more explicitly in the revision and highlight broader validation as an important direction for future work.
>
> To further address robustness, we have added new ablations with weight decay and QKNorm. Because the weight decay term is scaled by the learning rate in the update rule, and Adam commonly uses a default value of 0.1, we adjusted the weight decay values for the other optimizers accordingly. The results are shown in Figures 1–4 at the anonymous link below.
>
> These additional experiments show that our main qualitative findings remain unchanged when weight decay is introduced. In particular, under QKNorm with weight decay, $p=1/4$ achieves the best validation loss (2.800), compared with 2.802 for $p=1/2$, 2.806 for $p=0$, and 2.815 for $p=1$. Thus, the relative advantage of intermediate spectral compression remains consistent in this setting as well. We will include these results in the revised version.
>
> [1] Adaptive Optimization via Momentum on Variance-Normalized Gradients
>
> [2] Adam Improves Muon: Adaptive Moment Estimation with Orthogonalized Momentum
>
> ---
>
> **Q2: Order of Transformations**
> > Why is $\Psi_p$ applied to $O_t^\text{rms}$ rather than applying the transformations in the reverse order?
>
> **Response:** Thank you for the question. Our goal is to isolate the optimization effect of spectral compression itself, so we apply $\Psi_p$ to the matrix actually used for the update.
>
> If the order were reversed, i.e., we applied $\Psi_p$ first and then Adam-style RMS normalization, the later coordinatewise normalization would generally distort the spectral structure introduced by $\Psi_p$. This would make it harder to determine whether any effect came from spectral compression or from the later elementwise renormalization.
>
> Applying $\Psi_p$ directly to $O_t^{\mathrm{rms}}$ lets us test whether spectral compression still matters after Adam-style RMS normalization. This is also more consistent with Muon, which transforms first-moment momentum information.
>
>
> ---
>
> **Q3: Layer-Adaptive Compression Ratio $p$**
> > Given the observations in the paper, have the authors considered adapting $p$ during training or using different values of $p$ for different layers? If so, do they have insights into how the appropriate transformation might depend on layer type?
>
> **Response:** Thank you for this suggestion. We have not explored adaptive choices of $p$ in this paper, but we agree it is an interesting direction.
>
> A plausible hypothesis is that smaller $p$ (stronger compression) may help in less stable phases of training, where spectral compression has a stronger stabilizing effect. As training becomes more stable, larger $p$ may become preferable, since they preserve more anisotropy and are less likely to amplify low-signal or noisy directions.
>
> A similar idea may apply across layers: layers with more anisotropic or unstable updates may benefit from smaller $p$, while better-conditioned layers may prefer values closer to the identity transform. We view this as an interesting direction for future work.
>
> ---
> [1] [Anonymous link](https://shorturl.at/beJnd)

---

> > ### Author Rebuttal · Reviewer_k39R · 2026-04-01
> >
> > Thank you for addressing my questions. I maintain my view that the paper is solid and will keep my score unchanged.

---

### Official Review · Reviewer_KSqo · 2026-03-09

**Soundness:** 3
**Presentation:** 3
**Significance:** 3
**Originality:** 3
**Overall Recommendation:** 4
**Confidence:** 3

**Summary:**

This paper studies the recently proposed Muon optimizer through a unified spectral lens. The core idea is to view Muon’s orthogonalized/polar update as the $p=0$ endpoint of a family of spectral transformations $\Psi_p(O)=U\Sigma^pV^\top$ applied to an update matrix $O$ (either momentum $M_t$ or RMS-normalized momentum $M_t\oslash \sqrt{V_t}$), and to evaluate intermediate variants $p\in \{1,\tfrac12,\tfrac14,0 \}$. To make fractional powers practical without SVD, the authors propose a coupled Newton–Schulz iteration to compute matrix roots/inverse-roots and thereby implement $\Psi_{1/2}$ and $\Psi_{1/4}$ using only matrix multiplications. Empirically (nanoGPT/GPT-2 124M on OpenWebText, controlled hyper-parameter protocol), they find: (i) on first-moment updates, spectral compression—especially full orthogonalization—improves stability versus momentum SGD; (ii) on RMS-normalized updates, Muon-style full flattening is not consistently better than Adam and can be worse than milder compression; and (iii) overall RMS-normalization dominates first-moment-only methods under their controlled setup.

**Compliance With Llm Reviewing Policy:**

Affirmed.

**Final Justification:**

The paper has clear strengths in framing Muon under a unified spectral perspective and in designing controlled experiments that isolate the effect of spectral transformations on matrix updates. I found the central message meaningful: Muon provides a stabilizing effect, especially for first-moment updates, but it does not consistently outperform Adam-style RMS-normalized methods. Overall, I viewed the paper as solid in soundness, originality, significance, and clarity.

My main concerns were about runtime overhead, Newton–Schulz implementation details, and numerical stability. The authors’ rebuttal addressed these concerns well by providing wall-clock results, clarifying the choice of $K=5$, discussing sensitivity to $K$, and explaining the convention behind $\kappa_0$ as well as the handling of rectangular matrices. These clarifications improved the reproducibility and practical interpretability of the work.

As a result, the rebuttal positively changed my evaluation. While I still think these implementation and cost details should be added clearly in the final version, and I believe the main concerns were adequately addressed.

**Key Questions For Authors:**

See above

**Limitations:**

- No runtime/compute accounting, despite potentially expensive matrix-function iterations.
- Unclear numerical safeguards for iterative matrix functions in the presence of ill-conditioning and rank issues.

**Strengths And Weaknesses:**

### Strengths

- Framing Muon as $p=0$ in $\Psi_p(O)=U\Sigma^pV^\top$ is conceptually clean and makes it easy to reason about "how much" spectral flattening is beneficial.
- The paper tries to isolate the effect of the matrix update rule by holding many factors fixed and explicitly separating matrix vs. vector parameter treatment.
- Given the hype around Muon, careful evidence that "Muon is stabilizing but not universally superior to Adam-style RMS methods" is a useful corrective.
- Avoiding explicit SVD via iterative matrix-function approximations is relevant if these methods are to be used at scale.

### Weaknesses

- **[W1]** Fractional spectral methods can be substantially more expensive than Adam's element-wise operations. Without wall-clock cost, memory overhead, or iteration count sensitivity, it’s hard to assess practical significance.
    - **[Q]** What is the wall-clock overhead of AdamS/AdamQ/mSGDQ relative to Adam/mSGD/Muon on your setup? How does overhead scale with matrix dimension?
- **[W2]** The number of Newton–Schulz iterations $K$, stability safeguards, damping/epsilon, and numerical failure modes are not fully discussed. Adam equations are presented without bias correction/epsilon. It is unclear what exactly is used in code.
    - **[Q]** What $K$ NS-iterations do you use for Muon’s polar iteration, coupled Newton–Schulz for $X^{\pm 1/2}$, and the two-stage method for $X^{\pm 1/4}$? How sensitive are results to $K$?
- **[W3]** You write $\kappa_0=1$. This is only true if $\kappa$ is defined over nonzero singular values (or if you assume full rank). For rank-deficient matrices, $\sigma_{\min}=0$ and the usual condition number is infinite. Please clarify your convention.
    - **[Q]** Under what conditions do the coupled iterations fail or become unstable during training (e.g., poorly scaled $O^\top O$, near rank-deficiency)? Do you use damping, epsilon shifts, clipping, or re-scaling beyond Frobenius normalization?

- **Typos/Presentations**:
    - In the main text you assume $m\ge n$, while Appendix A states $m\le n$. Please clearly state and how rectangular cases are handled in implementation.
    - Figure 4 legend says "adamw-, sgdw-", but you state weight decay is disabled. Consider renaming to avoid confusion.

---

> ### Author Rebuttal · Authors · 2026-03-31
>
> We would like to thank the reviewer for the constructive feedbacks. We are encouraged that the reviewer found the spectral framing clean and the conclusions about Muon useful. Below we address the main concerns on computational cost, Newton-Schulz details, and scope.
>
> ---
> **Q1: Newton-Schulz Overhead**
> > **[W1]** Fractional spectral methods can be substantially more expensive than ...
> >
> > **[Q]** What is the wall-clock overhead of AdamS/AdamQ/mSGDQ relative to Adam/mSGD/Muon ...?
>
> **Response:** Thank you for this question. On our setup (sequence length 1024, batch size 480, 200k steps), the Newton-Schulz (NS) spectral variants introduce a moderate wall-clock overhead.
>
> On a single 8-GPU A800 node, the total training time for one full run is:
>
> |Optimizer|Time|
> |---|---|
> |Adam|33h05m|
> |AdamS ($p=1/2$)|34h18m|
> |AdamQ ($p=1/4$)|34h21m|
> |AdamZ ($p=0$)|34h09m|
>
> For $W\in\mathbb{R}^{m\times n}$ with $n\le m$, a $K$-step NS fractional transform has complexity $O(Kn^3+mn^2)$: the $Kn^3$ term comes from repeated operations on the smaller $n\times n$ matrix, while the $mn^2$ term comes from multiplying back with the original rectangular matrix. This cost is independent of batch size. The main additional memory overhead arises from temporary $n\times n$ matrices. We will include these wall-clock and scaling details in the revised version. Each layer has its own independent optimizer parameters, so the algorithm can be significantly accelerated through parallelization.
>
> ---
> **Q2: Reviewer's comments on the scale of experiments**
> > **[W2]** The number of Newton–Schulz iterations $K$, ..., are not fully discussed. ...
> >
> > **[Q]** What $K$ NS-iterations do you use for Muon’s polar iteration, coupled Newton–Schulz for $\mathbf{X}^{\pm 1/2}$, and ...?
>
>
> **Response:** Thank you for this question. In the Adam-family methods, we follow standard AdamW conventions, including bias correction and denominator $\epsilon=10^{-8}$. For NS, we use matrix normalization together with a small diagonal regularization term.
>
> For the spectral transforms, we use $K=5$ NS iterations for all the approximations in our main experiments. The $p=1/2$ case applies this routine once, whereas the $p=1/4$ case applies it twice. To evaluate the sensitivity to $K$, we compare the NS approximation with the exact SVD on Gaussian matrices of sizes $512\times512$ and $1024\times1024$, with condition numbers ranging from $1$ to $100$:
>
> |$K$|cos-sim ($p=1/2$)|cos-sim ($p=1/4$)|
> |---:|---:|---:|
> |$3$|$0.990$|$0.975$|
> |$5$|$0.997$|$0.993$|
> |$7$|$0.999$|$0.999$|
> |$10$|$1.000$|$1.000$|
>
> From these results, the NS approximation improves rapidly as $K$ increases, and at $K=5$, the results are already very strong. We will add these implementation details and the $K$-sensitivity results in the revision.
>
> ---
>
> **Q3: Coupled Iteration Failcases**
> > **[W3]** You write $\kappa_0 =1$. This is only true if $\kappa$ ...
> >
> > **[Q]**  Under what conditions do the coupled iterations fail or become unstable during training ...?
>
> **Response:** Thank you for this question. We would like to clarify that when we wrote $\kappa_0=1$, we meant the condition number computed over the nonzero singular values after the $p=0$ transform. For rank-deficient matrices, the standard condition number over all singular values is infinite, and we will make this point clear in the revision.
>
> The coupled iterations are most vulnerable when $O^\top O$ is poorly scaled or nearly singular. In our implementation, in addition to matrix normalization, we apply a small diagonal regularization / epsilon shift to stabilize the inverse-root computation. We do not rely on clipping within the NS routine itself.
>
> Empirically, this strategy is sufficient across all tested settings. Even for near-rank-deficient matrices with $\sigma_{\min}$ ranging from $10^{-3}$ down to $0$, the $p=1/2$ and $p=1/4$ approximations remain stable, with cosine similarity close to $1$. We will add this clarification together with a more detailed discussion of numerical stability in the revision.
>
> ---
> **Q4: Typos and Presentation**
> > In the main text you assume $m \ge n$, while ...
> > Figure 4 legend says "adamw-, sgdw-", ...
>
> **Response:** Thank you very much for pointing out these issues. In implementation, we always apply the NS routine to the smaller Gram matrix. For $X\in\mathbb{R}^{m\times n}$ with SVD $X=U\Sigma V^\top$:
>
> - If $m\ge n$, we use $S=X^\top X\in\mathbb{R}^{n\times n}$ and compute
>   $$
>   X S^{-1/4} = X (X^\top X)^{-1/4} = U\Sigma^{1/2}V^\top.
>   $$
> - If $m<n$, we use $S=XX^\top\in\mathbb{R}^{m\times m}$ and compute
>   $$
>   S^{-1/4} X = (XX^\top)^{-1/4}X = U\Sigma^{1/2}V^\top.
>   $$
>
> The same idea applies to the $p=0$ and $p=1/4$ cases. We will clarify this more explicitly in the revision.
> We will also revise  the legend in Figure 4 accordingly.
>
> ---
> [1] [Anonymous link](https://shorturl.at/beJnd)

---

> > ### Author Rebuttal · Reviewer_KSqo · 2026-04-02
> >
> > I would like to thank the authors for their thorough rebuttal. It has addressed most of my concerns, and I am glad to adjust my rating.

---

### Official Review · Reviewer_NSoy · 2026-03-13

**Soundness:** 3
**Presentation:** 3
**Significance:** 2
**Originality:** 2
**Overall Recommendation:** 4
**Confidence:** 3

**Summary:**

The paper studies Muon through a unified spectral family of update transformations, where Muon corresponds to full singular-value flattening and intermediate variants apply weaker spectral compression. The authors instantiate this family for both first-moment and RMS-normalized updates, derive coupled Newton–Schulz approximations for the fractional variants, and evaluate the resulting optimizers on 124M-parameter nanoGPT training on OpenWebText. The main empirical conclusion is that spectral compression strongly stabilizes first-moment updates, but once RMS normalization is present, Muon-style orthogonalization does not show a clear advantage over Adam-like updates and partial compression can perform similarly or slightly better.

**Compliance With Llm Reviewing Policy:**

Affirmed.

**Final Justification:**

The paper’s main strength is that it provides a clean and useful controlled perspective on Muon through a unified spectral-family formulation. I find this framing conceptually valuable and the resulting empirical message meaningful: spectral compression appears to act primarily as a stabilizer for first-moment matrix updates, while offering more limited benefit once RMS normalization is already present. In that sense, the paper makes a worthwhile diagnostic contribution even if it does not introduce a definitively superior optimizer. The presentation is generally clear, the scope is now better articulated, and the authors are appropriately cautious in their claims.

My initial concerns were mainly about the practical relevance and completeness of the evaluation: the narrow experimental regime, the absence of weight-decay ablations, the lack of analysis of the Newton–Schulz approximation, and the gap to more modern practice. The rebuttal addressed a substantial part of these concerns. In particular, the additional evidence and clarification around weight decay / QKNorm and the Newton–Schulz approximation materially improved my assessment of the paper’s soundness and reproducibility, and I appreciate the authors’ effort to clarify that the paper is intended as a controlled analysis rather than a fully practice-oriented benchmark.

Some weaknesses remain. I still think an explicit learning-rate decay ablation would be important, since LR decay can materially affect optimizer behavior and potentially alter the relative ranking of methods; a batch-size ablation would also strengthen the practical significance of the conclusions. More broadly, the evaluation remains limited in scope, so I do not view the paper as a definitive statement about Muon in modern large-scale practice. These concerns temper my enthusiasm and keep my assessment closer to weak accept than a stronger endorsement.

Overall, however, the rebuttal positively changed my evaluation. I find the core perspective useful, the controlled study worthwhile, and the revised paper likely to be of interest to researchers working on optimizer design and understanding.

**Key Questions For Authors:**

1. Why were the spectral powers $p \in \\{1, 1/2, 1/4, 0\\}$ chosen? Is there theoretical, empirical, or practical motivation for these specific values? A stronger justification would improve my view of the paper.

2. Can you provide at least one ablation with non-zero weight decay and, ideally, a more modern Muon configuration with matched update norms / scale tuning? This would substantially affect my assessment of practical significance.

3. How robust are the reported differences across random seeds? Please provide variance estimates or multi-seed results, especially for the small gaps between Muon and the RMS-normalized variants. This would affect my assessment of soundness.

4. Can you repeat the key experiments at one substantially different batch size, or explain more carefully why batch size 480 is sufficient for the conclusions you draw? This matters because optimizer comparisons are known to depend strongly on batch size.

5. Please include an ablation on the Newton–Schulz approximation itself, including iteration count, approximation quality, and runtime overhead. This would improve reproducibility and clarify the computational tradeoffs.

**Limitations:**

Partially. The paper does acknowledge some limitations, but this discussion should be expanded. In particular, the authors should discuss more explicitly the lack of learning-rate decay, the single batch size / model scale, the absence of statistical uncertainty estimates, and the lack of ablations on the Newton–Schulz approximation. A brief but more specific discussion of broader practical implications would also improve this section.

**Strengths And Weaknesses:**

I find the paper’s central idea interesting and potentially useful. A controlled dissection of Muon is valuable, and the spectral-family view is a clean way to compare full orthogonalization, partial compression, and the identity transformation. I also appreciate that the paper does not overclaim and instead arrives at a more nuanced conclusion than much of the surrounding discussion.

**Soundness.** The study is informative but currently too limited to support some of its broader implications. The experimental setup is narrow: one model scale, one relatively small batch size, no weight decay, and apparently no learning-rate decay beyond warmup. This is a substantial gap from modern practice, especially for Muon, where prior work suggests that weight decay, update-scale matching, and related details matter materially. Because of this, I do not think the current experiments are sufficient to support strong practical conclusions about Muon versus Adam-like methods. In addition, I did not see repeated-seed results or uncertainty estimates, so it is hard to judge whether the smaller reported differences are statistically meaningful.

**Presentation.** The paper is readable overall, but several points need clarification. The motivation for choosing exactly the powers $p \in \\{1, 1/2, 1/4, 0\\}$ is not convincing enough. The discussion of QK-Norm / QK-Clip in the introduction is also too vague and should either be substantiated more clearly or softened. The related-work section misses several relevant prior directions in matrix and spectral optimization, including older matrix optimizers and more recent work on improved polar-factor approximations, as well as adaptive Muon variants such as AdaMuon / NorMuon-like hybrids. Some presentation details also hurt readability: the token budget should be stated explicitly, Figure 2 would be easier to compare with shared y-axes, and the running title / terminology should be cleaned up.

**Significance.** I currently see the paper as potentially interesting but somewhat limited in impact in its present form. The controlled finding that spectral compression mainly helps stabilize unnormalized first-moment updates, while offering limited benefit once RMS normalization is already present, is useful. However, the lack of important ablations and the gap to modern training practice reduce the practical value of the conclusions. I am open to changing my mind if the authors can show that the main observations persist under more realistic settings.

**Originality.** I think the main contribution is the unified spectral framing and the controlled comparison it enables. That is a worthwhile perspective. However, the computational aspect feels more incremental than the paper currently suggests, and the novelty claim would be stronger with better positioning against prior work on spectral / matrix optimizers and improved orthogonalization approximations.

Overall, I find the idea promising and the controlled study worthwhile, but the paper currently feels more like an interesting diagnostic analysis than a definitive evaluation. The missing discussions and ablations materially limit my enthusiasm.

---

> ### Author Rebuttal · Authors · 2026-03-31
>
> We thank the reviewer for the thoughtful feedbacks. We are encouraged that the reviewer found the unified spectral perspective clean, controlled, and useful.
>
> We hope the responses below address your concerns.
>
> ---
> **Q1: Spectral Power Choice**
> > Why were the spectral powers $p\in \{1, 1/2, 1/4, 0\}$ chosen?
>
> **Response:** Thank you very much for the question. We chose $p=1/2$ and $p=1/4$ because they can be computed efficiently via matrix functions without requiring an explicit SVD. In particular, we use the coupled Newton-Schulz iteration (Sec. 3.5; Appendix C/D) to compute these cases efficiently. In contrast, other exponents such as $p=1/3$ are substantially more difficult to compute accurately without resorting to SVD.
>
> ---
> **Q2: Weight Decay and Muon Configuration**
> > Can you provide at least one ablation with non-zero weight decay and, ideally, a more modern Muon ...?
>
> **Response:** To address this concern, we added ablation studies with weight decay and QKNorm. Since the weight decay term is multiplied by the learning rate in the parameter update, and Adam commonly uses a default value of 0.1, we scaled the weight decay values accordingly for the other optimizers. The corresponding results are shown in Figures 1-4 at the anonymous link below.
>
> QKNorm and weight decay consistently improve model performance. We observe that $p=1/4$ achieves the best validation loss of 2.800, compared with 2.802 for $p=1/2$, 2.806 for $p=0$, and 2.815 for $p=1$.
>
> These experiments suggest that our main qualitative findings remain largely unchanged when weight decay is introduced. We will include these results in the revised version.
>
> ---
> **Q3: Random Seed Variance Estimates**
> > How robust are the reported differences across random seeds? ... This would affect my assessment of soundness.
>
> **Response:** Our models were trained on 100B tokens, and the randomness for different seeds is very small. According to our previous extensive experiments, the variance between different random seeds is smaller than 0.001.
>
> ---
> **Q4: Batch Size Selection**
> > Can you repeat the key experiments at one substantially different batch size, ..., strongly on batch size.
>
> **Response:** Our models were trained with 100B tokens, with each iteration using around 500K tokens. The batch size is large. This setting, consistent with standard nanoGPT setups, is also used in several recent nanoGPT-style studies, including [1,2].
>
> We would like to clarify that our goal is to isolate spectral effects in a controlled and commonly used GPT-2 124M / OpenWebText regime (context length 1024, batch size 480). Considering the large batch size and number of tokens used in this work, the randomness in batch size is small.
>
> [1] Adaptive Optimization via Momentum on Variance-Normalized Gradients
>
> [2] Adam Improves Muon: Adaptive Moment Estimation with Orthogonalized Momentum
>
> ---
> **Q5: Newton-Schulz Details**
> > Please include an ablation on the Newton–Schulz approximation itself, including iteration count, approximation quality, and runtime overhead.
>
> **Response:** Thank you for the suggestion. To address this concern, we now include an ablation study of the NS approximation, covering the iteration count, approximation quality, and runtime overhead.
>
> In terms of runtime, NS introduces a moderate overhead. On 8 A800 GPUs, a 200k-step training run takes:
>
> |Optimizer|Time|
> |---|---|
> |Adam|33h05m|
> |AdamS ($p=1/2$)|34h18m|
> |AdamQ ($p=1/4$)|34h21m|
> |AdamZ ($p=0$)|34h09m|
>
> Each layer has its own independent optimizer parameters, so the algorithm can be significantly accelerated through parallelization.
>
> To evaluate approximation quality, we generate Gaussian matrices of sizes $512\times512$ and $1024\times1024$ with condition numbers from 1 to 100, and compare the NS approximation against the exact SVD solution. Representative results are:
>
> |$K$|cos-sim ($p=1/2$)|cos-sim ($p=1/4$)|
> |---:|---:|---:|
> |$3$|$0.990$|$0.975$|
> |$5$|$0.997$|$0.993$|
> |$7$|$0.999$|$0.999$|
> |$10$|$1.000$|$1.000$|
>
> The approximation improves rapidly as $K$ increases, and by $K=5$, both methods have already achieved very high cosine similarity. Therefore, we use $K=5$ in our main experiments.
>
> We also conducted experiments with $K=5$ and $K=10$, shown in Figure 5 at the anonymous link. We observe that their training performance is nearly identical.
>
> ---
> **Q6: Broader Practical Limitations**
> > The paper does acknowledge some limitations, but this discussion should be expanded. In particular, ..., improve this section.
>
> **Response:** We would like to clarify that our primary focus is to compare different optimizers under a controlled experimental setting. During the rebuttal stage, we added additional ablation studies on weight decay and QKNorm. Other factors, including learning-rate decay, additional batch-size and model-scale settings, and the quality-efficiency tradeoff of the NS approximation, are all important directions for future work.
>
> ---
> [1] [Anonymous link](https://shorturl.at/beJnd)

---

> > ### Author Rebuttal · Reviewer_NSoy · 2026-04-04
> >
> > I thank the authors for the detailed rebuttal and additional experimental clarifications. The response largely addresses my main concerns. In particular, the added discussion and ablations around weight decay / QKNorm and the Newton–Schulz approximation improve the paper materially, and I appreciate the authors’ effort to clarify the intended scope of the study as a controlled analysis rather than a fully practice-oriented benchmark.
> >
> > I still think that an **ablation with learning-rate decay is needed**, as LR decay can have a substantial effect on optimization outcomes and may even potentially change the relative ranking of methods; similarly, a batch-size (more optional) ablation would further strengthen the practical relevance of the conclusions. However, these now seem to me more like limitations of scope than fatal flaws.
> >
> > Overall, the rebuttal has positively updated my view of the paper. I find the core perspective useful and the controlled study worthwhile, and I would be willing to increase my score accordingly.
> >
> > ---
> > # Updated
> >
> > To clarify my previous acknowledgement, my updated view should not be read as unconditional support for acceptance. The rebuttal has positively changed my assessment, but my willingness to support acceptance is conditional on including the LR-decay ablation in the final version.

---

> > > ### Author Response · Authors · 2026-04-07
> > >
> > > Dear Reviewer NSoy,
> > >
> > > &nbsp;
> > >
> > >    Thank you very much for your positive feedback. We are glad that our clarification addressed your main concerns.
> > >
> > >    We really appreciate your time and consideration, and we are happy to know you are willing to raise your score.
> > >
> > >    __At this moment, it seems the score in the system has not yet been updated.  Can you please update the system?__
> > >
> > > &nbsp;
> > >
> > > Best Regards,
> > > Authors

---

### Decision · Program_Chairs · 2026-04-30

**Decision:**

Accept (spotlight)

**Comment:**

The reviewers are overall positive on the paper and find the main contribution meaningful. In particular, they view the spectral reframing of Muon within a broader family of spectral transformations as conceptually useful, and the controlled empirical study as informative. The rebuttal addressed most of the main concerns, especially those related to runtime overhead, Newton–Schulz implementation details, numerical stability, and added ablations with weight decay and QKNorm.

The main remaining concern is one of scope rather than correctness. In particular, the evaluation remains limited to a single controlled setup, and Reviewer NSoy noted that a learning-rate decay ablation is still important, as it could affect the relative ranking of methods. More broadly, the paper remains closer to a controlled analysis than to a broad practical benchmark. I suggest to the author to include such a comparison or a discussion.

Overall, the paper is viewed as technically solid, with the remaining weaknesses mainly concerning breadth of evaluation rather than the validity of the central contribution.